# Easier Painting Than Thinking: Can Text-to-Image Models Set the Stage, but Not Direct the Play?

**Ouxiang Li**[1][*], **Yuan Wang**[1], **Xinting Hu**[1][†], **Huijuan Huang**[2][‡], **Rui Chen**[2], **Jiarong Ou**[2],
**Xin Tao**[2][†], **Pengfei Wan**[2], **Xiaojuan Qi**[3], **Fuli Feng**[1]

[1]University of Science and Technology of China, [2]Kling Team, Kuaishou Technology, [3]The University of Hong Kong
lioox@mail.ustc.edu.cn, joyhu1412@gmail.com, jiangsutx@gmail.com

🌐 **Homepage** 🏆 **Leaderboard** ⚙ **Code** 🤗 **Benchmark** 🖼 **Dataset**

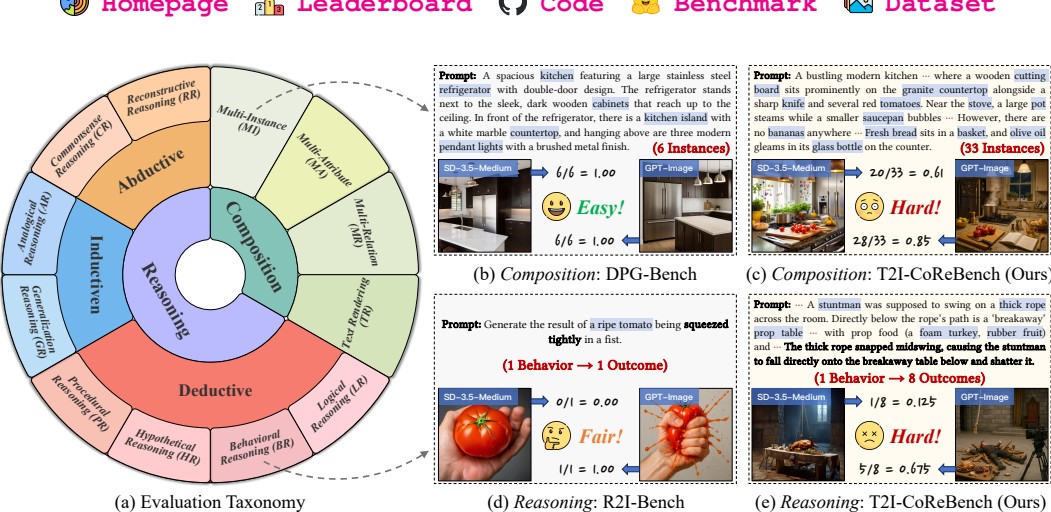

Figure 1: **Overview of our T2I-CoReBench**. (a) Our benchmark comprehensively covers two fundamental T2I capabilities (*i.e.*, *composition* and *reasoning*), further refined into 12 dimensions. (b-e) Our benchmark poses greater challenges to advanced T2I models, with higher compositional density than DPG-Bench (Hu et al., 2024) and greater reasoning intensity than R2I-Bench (Chen et al., 2025b), enabling clearer performance differentiation across models under real-world complexities. Each image is scored based on the ratio of correctly generated elements.

## Abstract

Text-to-image (T2I) generation aims to synthesize images from textual prompts, which jointly specify what must be shown and imply what can be inferred, which thus correspond to two core capabilities: *composition* and *reasoning*. Despite recent advances of T2I models in both composition and reasoning, existing benchmarks remain limited in evaluation. They not only fail to provide comprehensive coverage across and within both capabilities, but also largely restrict evaluation to low scene density and simple one-to-one reasoning. To address these limitations, we propose **T2I-CoReBench**, a comprehensive and complex benchmark that evaluates both composition and reasoning capabilities of T2I models. To ensure comprehensiveness, we structure composition around scene graph elements (*instance*, *attribute*, and *relation*) and reasoning around the philosophical framework of inference (*deductive*, *inductive*, and *abductive*), formulating a 12-dimensional evaluation taxonomy. To increase complexity, driven by the inherent real-world complexities, we curate each prompt with higher compositional density for composition and greater reasoning intensity for reasoning. To facilitate fine-grained and reliable evaluation, we also pair each evaluation prompt with a checklist that

---

[*]Work done during internship at Kling Team, Kuaishou Technology.
[†]Corresponding authors.
[‡]Project lead.

specifies individual *yes/no* questions to assess each intended element independently. In statistics, our benchmark comprises $1,080$ challenging prompts and around $13,500$ checklist questions. Experiments across 38 current T2I models reveal that their composition capability still remains limited in high compositional scenarios, while the reasoning capability lags even further behind as a critical bottleneck, with all models struggling to infer implicit elements from prompts.

# 1 INTRODUCTION

Recent developments in text-to-image (T2I) generative models are advancing toward high-quality image generation that adheres to user instructions. In real-world applications, textual prompts are usually concise yet underspecified (Hutchinson et al., 2022; Zhong et al., 2023), conveying not only explicit descriptions about what must be depicted, but also implicit contextual cues for generating coherent and plausible images. These correspond to two fundamental capabilities required for faithful T2I generation: *composition* and *reasoning*. As shown in Fig. 1, *composition* aims to correctly generate all explicit visual elements in the prompt, including instances (*e.g.*, *tomato*), attributes (*e.g.*, *wooden*), and relations (*e.g.*, *next to*); *reasoning* aims to generate visual elements implicitly inferred from the prompt (*e.g.*, *a ripe tomato is squeezed tightly in a fist* → *the tomato juice bursts out*).

Predominant T2I models, primarily based on diffusion (Ho et al., 2020; Ho & Salimans, 2022; Peebles & Xie, 2023) and autoregressive paradigms (Sun et al., 2024; Li et al., 2024b), demonstrate strong performance on simple compositional tasks (Huang et al., 2023a; Ghosh et al., 2023) but still struggle with complex compositional tasks involving multiple visual elements (Hu et al., 2024; Wu et al., 2024) as well as reasoning tasks (Niu et al., 2025; Chen et al., 2025b). Recently, T2I models enhanced with large language models (LLMs) or multimodal LLMs (MLLMs) (Chameleon, 2024; Xie et al., 2024; Deng et al., 2025a; Wu et al., 2025a; Xu et al., 2025a;b) have emerged, which offer stronger text modeling and cross-modal alignment. This paradigm brings new expectations to handle more complex scenarios involving high compositional density and reasoning intensity.

Given these developments and challenges, it is increasingly important to establish a fair and holistic evaluation of T2I models that systematically assesses both composition and reasoning capabilities. Early efforts (Huang et al., 2023a; Ghosh et al., 2023; Li et al., 2024a) focus on evaluating basic composition capabilities with a limited number of visual elements. Subsequent benchmarks further extend the number of visual elements in composition (see Fig. 1 (b)) (Hu et al., 2024; Wu et al., 2024; Zhou et al., 2025) and evaluate certain reasoning capabilities (*e.g.*, behavioral reasoning in Fig. 1 (d)) (Fu et al., 2024; Niu et al., 2025; Chen et al., 2025b). These existing benchmarks exhibit two limitations. *(1) Lack of comprehensiveness:* Most benchmarks focus on either composition or reasoning in isolation, and their underlying taxonomies are largely heuristic, which prevents them from systematically capturing all relevant evaluation dimensions. *(2) Lack of complexity:* While some benchmarks increase the number of visual elements in composition, they remain limited to low scene density and fail to reflect the compositional complexity of real-world applications (*e.g.*, *generate a bustling modern kitchen* in Fig. 1 (c)). More importantly, current reasoning-oriented benchmarks mainly target single-step inference (*e.g.*, one behavior → one outcome), thus overlooking the multi-step causal chains inherent to real-world scenarios (see Fig. 1 (e)).

To address the above limitations, we introduce **T2I-COREBENCH**, a **Co**mposition and **Re**asoning **Bench**mark for systematic evaluation of T2I models. ***To ensure comprehensiveness,*** as illustrated in Fig. 1 (a), our taxonomy jointly covers composition and reasoning. For composition, we follow the scene graph structure (Johnson et al., 2015; Chang et al., 2021) and define three basic dimensions to fully depict a compositional scene: *instance*, *attribute*, and *relation*. We also include *text rendering* to capture the unique challenges of generating texts with precise content and layout. For reasoning, we adopt a tripartite framework of *deductive*, *inductive*, and *abductive* reasoning, as well-established in philosophical literature (Peirce, 1934; Zalta et al., 2003; Godfrey-Smith, 2009), and refine it into eight dimensions tailored to T2I scenarios. ***To increase complexity,*** as summarized in Table 1, we design each dimension with higher compositional density and increased reasoning difficulties compared with existing benchmarks. For composition, we increase the number of visual elements ($\sim 20$ per prompt) to simulate semantically dense scenarios. For reasoning, complexity is introduced along one-to-many (*i.e.*, one behavior → multiple outcomes) and many-to-one (*e.g.*, multiple premises → one conclusion) inferences, reflecting the intricate reasoning patterns in real-world applications.

Table 1: **T2I benchmark comparison.** Our T2I-COREBENCH comprehensively covers 12 evaluation dimensions spanning both *composition* ( **MI** *Multi-Instance*, **MA** *Multi-Attribute*, **MR** *Multi-Relation*, **TR** *Text Rendering*) and *reasoning* ( **LR** *Logical Reasoning*, **BR** *Behavioral Reasoning*, **HR** *Hypothetical Reasoning*, **PR** *Procedural Reasoning*, **GR** *Generalization Reasoning*, **AR** *Analogical Reasoning*, **CR** *Commonsense Reasoning*, and **RR** *Reconstructive Reasoning*). The symbols denote different coverage levels: ● indicates high compositional (visual elements $> 5$) or reasoning (one-to-many or many-to-one inference) complexity, ◐ indicates simple settings (visual elements $\leq 5$ or one-to-one inference), and ○ indicates no coverage.

| Benchmark | Composition | | | | Reasoning | | | | | | | |
| | | | | | Deductive | | | | Inductive | | Abductive | |
| | MI | MA | MR | TR | LR | BR | HR | PR | GR | AR | CR | RR |
|---|---|---|---|---|---|---|---|---|---|---|---|---|
| T2I-CompBench (Huang et al., 2023a) | ◐ | ◐ | ◐ | ○ | ○ | ○ | ○ | ○ | ○ | ○ | ○ | ○ |
| GenEval (Ghosh et al., 2023) | ◐ | ◐ | ◐ | ○ | ○ | ○ | ○ | ○ | ○ | ○ | ○ | ○ |
| GenAI-Bench (Li et al., 2024a) | ◐ | ◐ | ◐ | ○ | ○ | ○ | ○ | ○ | ○ | ○ | ○ | ○ |
| DPG-Bench (Hu et al., 2024) | ● | ● | ● | ○ | ○ | ○ | ○ | ○ | ○ | ○ | ○ | ○ |
| ConceptMix (Wu et al., 2024) | ◐ | ◐ | ◐ | ○ | ○ | ○ | ○ | ○ | ○ | ○ | ○ | ○ |
| TIIF-Bench (Wei et al., 2025) | ◐ | ◐ | ◐ | ○ | ○ | ○ | ○ | ○ | ○ | ○ | ○ | ○ |
| LongBench-T2I (Zhou et al., 2025) | ● | ● | ● | ○ | ○ | ○ | ○ | ○ | ○ | ○ | ○ | ○ |
| PRISM-Bench (Fang et al., 2025) | ● | ◐ | ◐ | ◐ | ○ | ○ | ○ | ○ | ○ | ○ | ○ | ○ |
| UniGenBench (Wang et al., 2025) | ◐ | ◐ | ◐ | ◐ | ◐ | ○ | ○ | ○ | ○ | ○ | ◐ | ○ |
| Commonsense-T2I (Fu et al., 2024) | ○ | ○ | ○ | ○ | ○ | ○ | ○ | ○ | ○ | ○ | ◐ | ○ |
| PhyBench (Meng et al., 2024) | ○ | ○ | ○ | ○ | ○ | ◐ | ○ | ○ | ○ | ○ | ◐ | ○ |
| WISE (Niu et al., 2025) | ○ | ○ | ○ | ○ | ○ | ○ | ○ | ○ | ○ | ○ | ◐ | ○ |
| T2I-ReasonBench (Sun et al., 2025) | ○ | ○ | ○ | ◐ | ○ | ○ | ○ | ○ | ○ | ○ | ◐ | ○ |
| R2I-Bench (Chen et al., 2025b) | ○ | ○ | ◐ | ○ | ◐ | ◐ | ◐ | ○ | ○ | ○ | ◐ | ◐ |
| OneIG-Bench (Chang et al., 2025) | ● | ● | ● | ● | ○ | ○ | ○ | ○ | ○ | ○ | ◐ | ○ |
| **T2I-COREBENCH (Ours)** | ● | ● | ● | ● | ● | ● | ● | ● | ● | ● | ● | ● |

To enable fine-grained and reliable evaluation, we pair each textual prompt with a checklist of independent *yes/no* questions, assessing whether the generated image faithfully captures both explicit and implicit visual elements. The generated images are then evaluated against these checklists by Gemini 2.5 Flash (Gemini Team, 2023), an MLLM evaluator selected for its strong alignment with human judgments and efficiency at scale. In total, T2I-COREBENCH encompasses 12 well-defined dimensions, with $1,080$ challenging prompts and approximately $13,500$ checklist questions. In experiments, we benchmark 38 current T2I models across architectures and scales, including diffusion models, autoregressive models, and unified models. Our study shows that composition capability in T2I generation is steadily improving, with open-source models gradually narrowing the gap with closed-source counterparts, whereas the overall performance remains inadequate in high compositional scenarios. Most notably, reasoning capability lags significantly behind: even the state-of-the-art (SOTA) models fail to reliably infer implicit visual elements from prompts, making reasoning the central bottleneck for advancing T2I generation. Our contributions can be concluded as follows:

- We introduce T2I-COREBENCH, the first benchmark that jointly emphasizes comprehensiveness and complexity in T2I evaluation, covering both composition and reasoning capabilities through $1,080$ challenging prompts across 12 dimensions.

- We pair each prompt with a human-verified checklist of individual *yes/no* questions, for a total of around $13,500$ questions across the benchmark. This facilitates fine-grained and reliable assessment of whether the generated images faithfully capture both explicit and implicit elements.

- We conduct comprehensive evaluations on 38 current T2I models and conclude valuable insights, revealing that composition, though steadily improving, still remains unsolved in complex scenarios, whereas reasoning lags markedly behind and stands as the central bottleneck.

## 2 RELATED WORKS

**Text-to-Image Generative Models.** In recent years, T2I generation has witnessed significant advancements, with its rapid development largely driven by the emergence of diffusion models (Ho

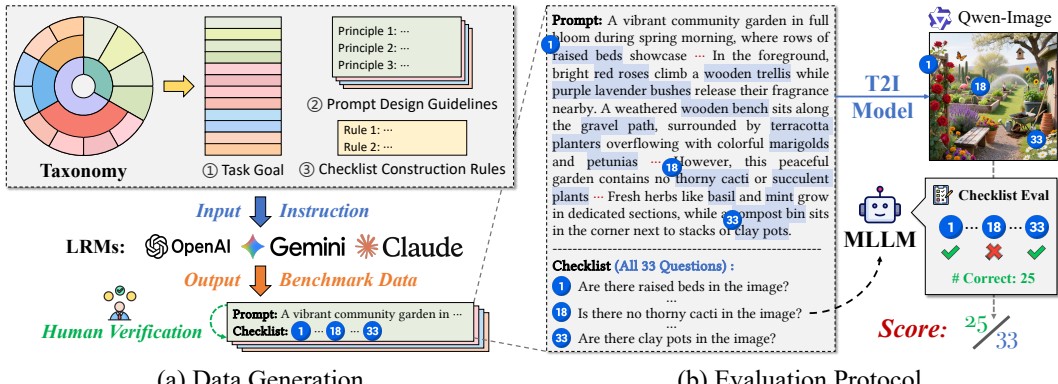

(a) Data Generation        (b) Evaluation Protocol

Figure 2: **Overview of our T2I-COREBENCH pipeline.**

et al., 2020; Ho & Salimans, 2022; Rombach et al., 2022; Wang et al., 2024; 2026). Predominant models, including the Stable Diffusion series (Esser et al., 2024), the Flux series (Black Forest Labs, 2024), and the DALL·E series (Ramesh et al., 2021), have led to substantial improvements in compositional text-image alignment. To better align with the textual modality at the token level, autoregressive (Sun et al., 2024; Li et al., 2024b; Tian et al., 2024; Han et al., 2025) and unified models (Chameleon, 2024; Xie et al., 2024; Chen et al., 2025c; Deng et al., 2025a; Chen et al., 2025a; Wu et al., 2025a) have emerged in an LLM-like architecture, demonstrating remarkable performance in composition tasks as well as reasoning tasks due to their autoregressive paradigm. Meanwhile, some approaches (Guo et al., 2025b; Li et al., 2025b; Liao et al., 2025; Duan et al., 2025) are exploring integrating reasoning into T2I generation to handle more complex and controllable tasks.

**Text-to-Image Evaluation Benchmarks.** Driven by the explicit or implicit nature of T2I generation, which requires both *composition* and *reasoning*. Early T2I benchmarks (Huang et al., 2023a; Ghosh et al., 2023; Li et al., 2024a) primarily target composition tasks with explicit visual elements. Subsequent benchmarks (Hu et al., 2024; Wu et al., 2024; Wei et al., 2025; Zhou et al., 2025; Fang et al., 2025) complicate the prompt with more detailed visual elements, yet still fall short in capturing the real-world challenge of high compositional density. In parallel, reasoning-oriented benchmarks (Fu et al., 2024; Meng et al., 2024; Niu et al., 2025; Chen et al., 2025b; Chang et al., 2025; Wu et al., 2025c; Sun et al., 2025; Wang et al., 2025; Li et al., 2025a; 2026) are gaining prominence as T2I models progress in reasoning tasks, including reasoning dimensions such as commonsense, logical, and causality. However, they primarily focus on simple one-to-one inference, overlooking more complex multi-step reasoning prevalent in real-world scenarios. Furthermore, their taxonomy of both capabilities is mostly heuristic, thereby failing to cover all relevant dimensions in evaluation.

## 3 T2I-COREBENCH

In this section, we introduce T2I-COREBENCH as shown in Fig. 2, a benchmark designed to evaluate both *composition* and *reasoning* capabilities under real-world complexities, including high compositional density and reasoning intensity. We first formulate a comprehensive T2I evaluation taxonomy with complexity specified for each dimension in Sec. 3.1. Building upon this taxonomy, we then outline the benchmark construction details in Sec. 3.2 and statistical analyses in Sec. 3.3.

### 3.1 EVALUATION DIMENSIONS

To address the limitations of previous benchmarks, which evaluate composition and reasoning in isolation using heuristic taxonomies, we formulate a comprehensive evaluation taxonomy that unifies both capabilities and reflects real-world generation challenges, as shown in Table 2.

**Composition.** Inspired by scene graph structures (Johnson et al., 2015; Chang et al., 2021), a visual scene (*e.g.*, an image) can be fully described by three components: instances, attributes, and relations. Based on this, we define three corresponding dimensions under real-world complexities, *i.e.*, **MI** *Multi-Instance*, **MA** *Multi-Attribute*, and **MR** *Multi-Relation*, to evaluate compositional capabil-

Table 2: **Definition of the 12 evaluation dimensions in our T2I-COREBENCH.** Each dimension is described with its definition, along with a complexity number that quantifies the **bolded** element, driven by the density of visual elements in composition and the intensity of inferences (one-to-many or many-to-one) in reasoning. More detailed descriptions can be found in Appx. A.1.

| | Dimension | Definition | #Complexity |
|---|---|---|---|
| **Composition** | **MI** *Multi-Instance* | *Generate multiple **instances** in a single image.* | $\sim 25$ |
| | **MA** *Multi-Attribute* | *Bind multiple **attributes** to a single subject.* | $\sim 20$ |
| | **MR** *Multi-Relation* | *Connect multiple **relations** within a unified scene.* | $\sim 15$ |
| | **TR** *Text Rendering* | *Render multiple **texts** with content fidelity and layout accuracy.* | $\sim 15$ |
| **Reasoning** | **LR** *Logical Reasoning* | *Solve **premise**-based puzzles through multi-step inference.* | $\sim 5$ |
| | **BR** *Behavioral Reasoning* | *Infer **visual outcomes** from initial states and subsequent behaviors.* | $\sim 8$ |
| | **HR** *Hypothetical Reasoning* | *Apply counterfactual premises and propagate their effects across **items**.* | $\sim 10$ |
| | **PR** *Procedural Reasoning* | *Reason over ordered multi-step **procedures** to derive the final scene.* | $\sim 5$ |
| | **GR** *Generalization Reasoning* | *Induce **rules** from examples and apply them to complete new scenes.* | $\sim 8$ |
| | **AR** *Analogical Reasoning* | *Transfer relational **rules** from a source domain to a target domain.* | $\sim 5$ |
| | **CR** *Commonsense Reasoning* | *Complete scenes by inferring unstated **commonsense elements**.* | $\sim 5$ |
| | **RR** *Reconstructive Reasoning* | *Reconstruct plausible initial states by tracing backward from **observed clues**.* | $\sim 5$ |

ities. Moreover, we introduce **TR** *Text Rendering* as a separate dimension to account for its unique complexity in content and layout accuracies of texts, as shown in Fig. 3 (a).

**Reasoning.** In T2I generation, prompts inevitably involve implicit visual elements, making reasoning a fundamental capability. To ensure a comprehensive evaluation, we adopt a tripartite framework of reasoning in philosophical literature (Peirce, 1934; Zalta et al., 2003; Godfrey-Smith, 2009), *i.e.*, *deductive*, *inductive*, and *abductive* reasoning. This framework provides a rigorous foundation for reasoning types, on which we define eight reasoning dimensions tailored to T2I scenarios.

- *Deductive Reasoning* is the process of drawing conclusions from given premises, ensuring that if the premises hold, the conclusion cannot be false. In T2I scenarios, this means generating images determined by the premises, based on which we define **LR** *Logical Reasoning*, **BR** *Behavioral Reasoning*, **HR** *Hypothetical Reasoning*, and **PR** *Procedural Reasoning*, as shown in Fig. 3 (b).

- *Inductive Reasoning* is the process of inferring conclusions from observed regularity patterns rather than from explicit premises. In T2I scenarios, this corresponds to inferring visual elements from underlying structural patterns in examples, based on which we define **GR** *Generalization Reasoning* and **AR** *Analogical Reasoning*, as shown in Fig. 3 (c).

- *Abductive Reasoning* is the process of reconstructing the most plausible explanation from observations. In T2I scenarios, this entails reconstructing hidden causes or unstated commonsense that best explain the visual observations, based on which we define **CR** *Commonsense Reasoning* and **RR** *Reconstructive Reasoning*, as shown in Fig. 3 (d).

## 3.2 BENCHMARK CONSTRUCTION

Building upon the evaluation dimensions defined in Sec. 3.1, we now construct T2I-COREBENCH through a standardized pipeline, as shown in Fig. 2. In our setup, each evaluation sample consists of a prompt, which guides T2I generation, and a checklist, which enables point-by-point verification of the generated visual elements. To systematically generate benchmark data across all dimensions, we design a unified instruction template, including: (1) *Task Goal*, outlining the evaluation objective of each dimension as described in Sec. 3.1; (2) *Prompt Design Guidelines*, specifying principles for constructing diverse and complex prompts as detailed in Sec. A.1; and (3) *Checklist Construction Rules*, defining how to decompose the target scene into atomic, objective, and verifiable questions. All samples undergo rigorous human verification to ensure quality and reliability in Appx. A.3.

**Prompt Design for Generation.** Since our benchmark features prompts with high compositional density and reasoning intensity, previous strategies prove inadequate: human-written prompts (Otani et al., 2023; Niu et al., 2025; Chang et al., 2025) are labor-intensive and lack scalability, while template-based prompts (Huang et al., 2023a; Ghosh et al., 2023; Wu et al., 2024) are rigid and limited in scene diversity. To overcome these issues, we leverage Large Reasoning Models (LRMs) to assist data construction, exploiting their broad knowledge to cover diverse scenes (Lee et al., 2023)

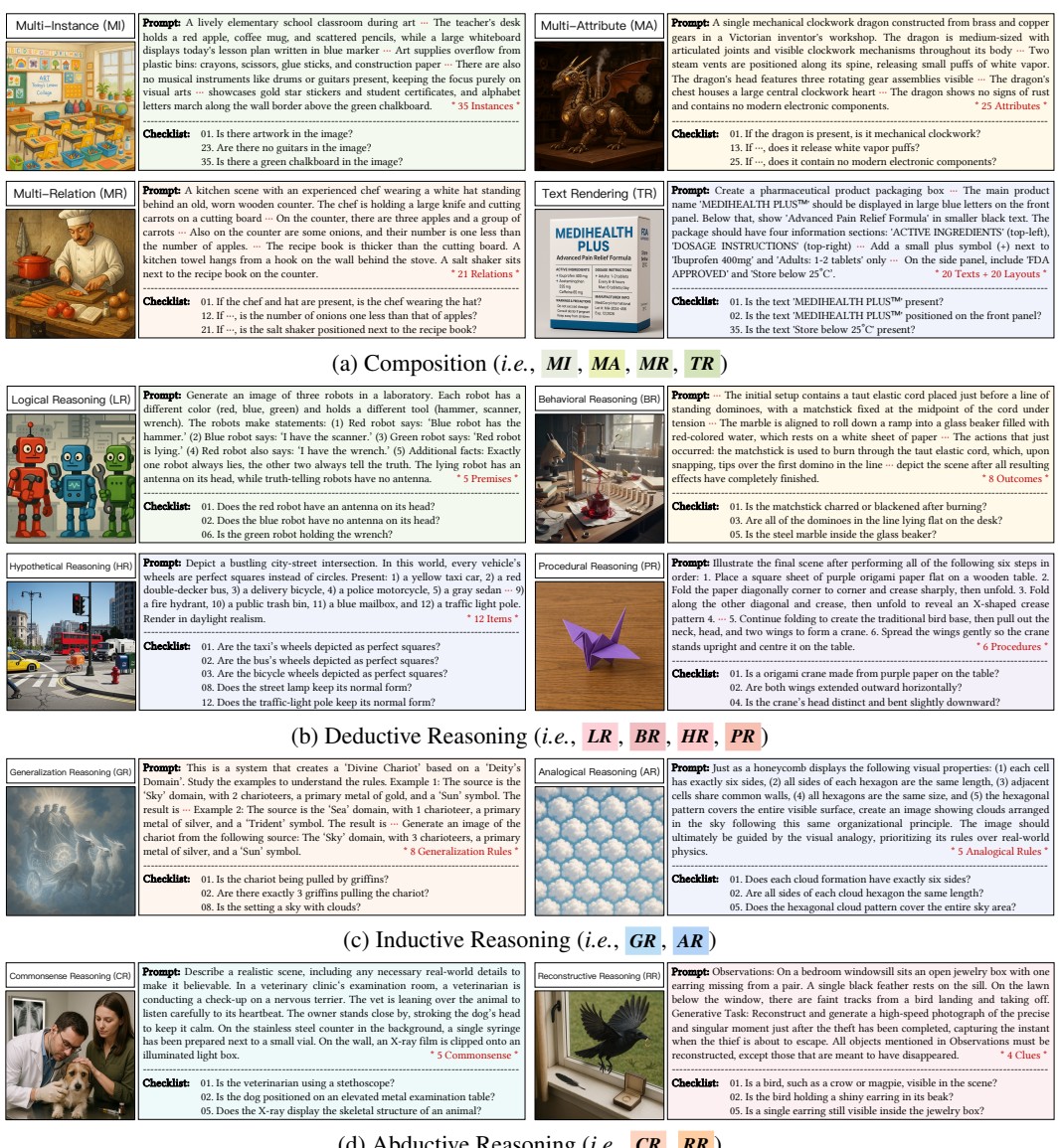

Figure 3: **Examples from T2I-CoReBench** illustrating (a) *composition* and (b-d) *reasoning* capabilities across 12 dimensions (see Appx. C.3 for complete versions). Each dimension is designed to incorporate *complexity* tailored to its unique characteristics, allowing more challenging evaluation under real-world scenarios, and supports fine-grained evaluation with human-verified checklists.

and strong reasoning capability to produce complex prompts (Zhong et al., 2024; Guo et al., 2025a). In practice, the *Prompt Design Guidelines* specify how to ensure sufficient diversity, semantic density, and reasoning complexity while keeping the prompt coherent, as detailed in Appx. A.2.

**Checklist Design for Evaluation.** Evaluating generations in complex scenarios requires more than existing metrics: (1) CLIPScore (Hessel et al., 2021) fails to account for multiple explicit elements and implicit reasoning outcomes; and (2) direct MLLM-based scoring (Li et al., 2024a) requires the model itself to infer intended outcomes with accumulated errors. To facilitate fine-grained and reliable evaluation of both explicit and implicit visual elements, we follow previous visual-question-answering–based evaluation paradigms (Hu et al., 2023; Yarom et al., 2023; Cho et al., 2023a;b), by pairing each prompt with a checklist of independent *yes/no* questions (with the correct answer always "*Yes*"). Specifically, we define a set of *Checklist Construction Rules* to decompose the target scene into atomic questions covering instances, attributes, relations, and reasoning outcomes in a verifiable manner, as detailed in Appx. A.2.

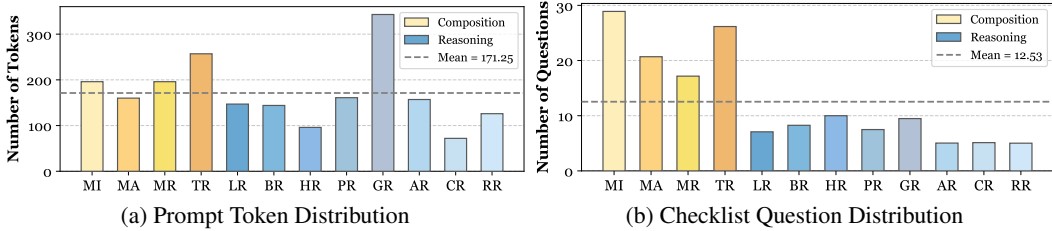

(a) Prompt Token Distribution  (b) Checklist Question Distribution

Figure 4: **Statistics of our T2I-COREBENCH** showing (a) prompt-token lengths and (b) checklist-question counts. Our benchmark exhibits high complexity in both *composition* and *reasoning* capabilities, with an average prompt length of 170 tokens and an average of 12.5 questions per sample.

**Evaluation Protocol.** Following previous protocols (Hu et al., 2023; 2024), we introduce an MLLM evaluator, *i.e.*, Gemini 2.5 Flash (Gemini Team, 2023), to conduct automatic evaluation by framing each item as a binary visual question answering task (*i.e.*, scored as "0" for "*no*" and "1" for "*yes*") in Fig. 2 (b). This protocol leverages the atomic checklist design, where each question targets an unambiguous visual element, ensuring inherent compatibility with MLLM-based evaluation.

## 3.3 STATISTICS AND ANALYSIS

To mitigate stylistic homogeneity and potential bias arising from relying on a single LRM (*e.g.*, using the same model to generate prompts and produce images often yields inflated performance since they share similar training data), we employ three SOTA LRMs for data construction, including Claude Sonnet 4 (Anthropic, 2025), Gemini 2.5 Pro (Gemini Team, 2023), and OpenAI o3 (OpenAI, 2025). In statistics, for each of the 12 evaluation dimensions, we collect 30 samples with each of the three LRMs, resulting in a total of 12 dimensions $\times$ 30 prompts $\times$ 3 LRMs $= 1,080$ generation prompts and $13,536$ questions in evaluation checklists, as detailed in Fig. 4.

## 4 EXPERIMENTS

### 4.1 EXPERIMENTAL SETUP

**Evaluated Models.** We evaluate 38 current T2I models across architectures and parameter scales, covering both open- and closed-models. The open-source pool includes 28 models: **(1) Diffusion Models:** SD-3-Medium, SD-3.5-Medium, SD-3.5-Large (Esser et al., 2024), FLUX.1-schnell, FLUX.1-dev, FLUX.1-Krea-dev (Black Forest Labs, 2024), FLUX.2-dev, FLUX.2-klein-4B, FLUX.2-klein-9B (Black Forest Labs, 2025), PixArt-$\alpha$ (Chen et al., 2023), PixArt-$\Sigma$ (Chen et al., 2024), HiDream-I1 (Cai et al., 2025), Qwen-Image, Qwen-Image-2512 (Wu et al., 2025a), HunyuanImage-3.0 (Cao et al., 2025), Z-Image-Turbo (Z-Image Team, 2025), LongCat-Image (LongCat Team, 2025); **(2) Autoregressive Models:** Infinity-8B (Han et al., 2025), GoT-R1-7B (Duan et al., 2025); and **(3) Unified Models:** BAGEL, BAGEL w/ Think (Deng et al., 2025b), show-o2-1.5B, show-o2-7B (Xie et al., 2025), Janus-Pro-1B, Janus-Pro-7B (Chen et al., 2025c), BLIP3o-4B, BLIP3o-8B (Chen et al., 2025a), OmniGen2-7B (Wu et al., 2025b). We further include 10 **closed-source commercial models**, including: Seedream 3.0 (Gao et al., 2025), Seedream 4.0 (Seedream et al., 2025), Seedream 4.5 (ByteDance, 2025), Gemini 2.0 Flash, Nano Banana, Nano Banana Pro (Gemini Team, 2023), Imagen 4, Imagen 4 Ultra (Google, 2025), GPT-Image (OpenAI, 2025a) and GPT-Image-1.5 (OpenAI, 2025b).

**Evaluation Details.** To facilitate automatic evaluation, we introduce Gemini 2.5 Flash (Gemini Team, 2023) as the MLLM evaluator, which exhibits strong vision-language performance aligned with humans (see Appx. C.1) at relatively low cost. Considering the possible unavailability of closed-source APIs in the future, we also report evaluation results with the open-source MLLMs, including: Qwen2.5-VL-72B-Instruct (Bai et al., 2025b), Qwen3-VL-8B-Thinking, Qwen3-VL-32B-Thinking, and Qwen3-VL-30B-A3B-Thinking (Bai et al., 2025a). More comprehensive and up-to-date evaluation results with these MLLM evaluators are available on our 🏆 **Leaderboard**. In evaluation, we report the mean score across all samples within each dimension as its final score for that dimension. More details can be found in Appx. B.

Table 3: **Main results on our T2I-CoReBench** assessing both *composition* and *reasoning* capabilities evaluated by Gemini 2.5 Flash. **#Params.** indicates the number of parameters in the image generation component, *e.g.*, diffusion transformer (Peebles & Xie, 2023). The shaded *Mean* column denotes the mean score for each capability. The best and second-best results are marked in **bold** and underline for open- and closed-models, respectively. More comprehensive and up-to-date evaluation results with additional MLLM evaluators are available on our 🏆 Leaderboard.

| Model | #Params. | Composition | | | | | Reasoning | | | | | | | | | Overall |
|---|---|---|---|---|---|---|---|---|---|---|---|---|---|---|---|---|
| | | MI | MA | MR | TR | *Mean* | LR | BR | HR | PR | GR | AR | CR | RR | *Mean* | |
| *Diffusion Models* | | | | | | | | | | | | | | | | |
| SD-3-Medium | 2B | 59.1 | 57.9 | 35.4 | 9.5 | 40.4 | 22.1 | 21.1 | 35.3 | 51.0 | 37.4 | 47.3 | 35.0 | 27.1 | 34.5 | 36.5 |
| SD-3.5-Medium | 2B | 59.5 | 60.6 | 33.1 | 10.6 | 41.0 | 19.9 | 20.5 | 33.5 | 53.7 | 33.4 | 52.7 | 35.6 | 22.0 | 33.9 | 36.3 |
| SD-3.5-Large | 8B | 57.5 | 60.0 | 32.9 | 15.6 | 41.5 | 22.5 | 22.4 | 34.2 | 52.5 | 35.5 | 53.0 | 42.3 | 25.2 | 35.9 | 37.8 |
| FLUX.1-schnell | 12B | 65.4 | 63.1 | 47.6 | 22.4 | 49.6 | 25.0 | 25.1 | 40.9 | 64.7 | 47.6 | 54.0 | 39.6 | 22.9 | 40.0 | 43.2 |
| FLUX.1-dev | 12B | 58.6 | 60.3 | 44.1 | 31.1 | 48.6 | 24.8 | 23.0 | 36.0 | 61.8 | 42.4 | 57.2 | 36.3 | 30.3 | 39.0 | 42.2 |
| FLUX.1-Krea-dev | 12B | 70.7 | 71.1 | 53.2 | 28.9 | 56.0 | 30.3 | 26.1 | 44.5 | 70.6 | 50.5 | 57.5 | 46.3 | 28.7 | 44.3 | 48.2 |
| FLUX.2-dev | 32B | **89.5** | **83.4** | **72.7** | **93.3** | **84.7** | **48.4** | 33.5 | 53.4 | 83.1 | 62.3 | 64.3 | **62.1** | 26.9 | **54.2** | **64.4** |
| FLUX.2-klein-4B | 4B | 80.0 | 77.1 | 63.5 | 53.0 | 68.4 | 34.6 | 30.5 | 49.5 | 77.3 | 58.1 | 58.1 | 48.8 | 25.2 | 47.7 | 54.6 |
| FLUX.2-klein-9B | 9B | 85.8 | 80.6 | 68.0 | 77.7 | 78.0 | 43.4 | 32.9 | 52.4 | 80.6 | 60.2 | 60.8 | 60.4 | 25.1 | 52.0 | 60.6 |
| PixArt-$\alpha$ | 0.6B | 40.2 | 42.2 | 14.2 | 3.3 | 25.0 | 11.6 | 11.6 | 21.1 | 30.4 | 22.6 | 44.4 | 26.7 | 20.9 | 23.7 | 24.1 |
| PixArt-$\Sigma$ | 0.6B | 47.2 | 49.7 | 23.8 | 2.8 | 30.9 | 14.7 | 18.3 | 26.7 | 39.2 | 25.7 | 44.9 | 33.9 | 24.3 | 28.5 | 29.3 |
| HiDream-I1 | 17B | 62.5 | 62.0 | 42.9 | 33.9 | 50.3 | 34.2 | 24.5 | 40.9 | 53.2 | 34.2 | 50.3 | 46.1 | 31.7 | 39.4 | 43.0 |
| Qwen-Image | 20B | 81.4 | 79.6 | 65.6 | 85.5 | 78.0 | 41.1 | 32.2 | 48.2 | 75.1 | 56.5 | 53.3 | 61.9 | 26.4 | 49.3 | 58.9 |
| Qwen-Image-2512 | 20B | 88.5 | 82.5 | 71.9 | 91.9 | 83.7 | 42.7 | 34.6 | 53.6 | 82.0 | 62.0 | 57.2 | 60.3 | 21.7 | 51.7 | 62.4 |
| Z-Image-Turbo | 6B | 79.5 | 72.2 | 62.7 | 83.9 | 74.6 | 39.1 | 28.8 | 48.7 | 74.0 | 56.2 | 55.8 | 52.0 | 26.0 | 47.3 | 56.4 |
| LongCat-Image | 6B | 81.4 | 74.5 | 61.5 | 65.7 | 70.8 | 39.1 | **35.7** | 48.5 | 75.5 | **72.5** | 61.4 | 58.8 | **41.0** | 54.1 | 59.6 |
| *Autoregressive Models* | | | | | | | | | | | | | | | | |
| Infinity-8B | 8B | 63.9 | 63.4 | 47.5 | 10.8 | 46.4 | 28.6 | 25.9 | 42.9 | 62.6 | 47.3 | 59.2 | 46.9 | 24.6 | 42.3 | 43.6 |
| GoT-R1-7B | 7B | 48.8 | 55.6 | 32.9 | 6.1 | 35.8 | 22.1 | 19.2 | 31.3 | 49.2 | 34.8 | 46.2 | 32.1 | 14.6 | 31.2 | 32.7 |
| *Unified Models* | | | | | | | | | | | | | | | | |
| BAGEL | 14B | 64.9 | 65.2 | 45.8 | 9.7 | 46.4 | 23.4 | 21.9 | 33.0 | 51.6 | 31.2 | 50.4 | 32.4 | 29.3 | 34.1 | 38.2 |
| BAGEL w/ Think | 14B | 57.7 | 60.8 | 37.8 | 2.2 | 39.6 | 25.5 | 25.4 | 33.9 | 58.6 | 53.5 | 56.9 | 41.6 | 39.8 | 41.9 | 41.1 |
| show-o2-1.5B | 1.5B | 59.5 | 60.3 | 36.1 | 4.6 | 40.1 | 21.6 | 21.8 | 37.1 | 47.7 | 39.9 | 44.7 | 29.0 | 24.0 | 33.2 | 35.5 |
| show-o2-7B | 7B | 59.4 | 61.8 | 38.1 | 2.2 | 40.4 | 23.2 | 23.1 | 37.5 | 51.6 | 40.9 | 47.2 | 32.2 | 21.3 | 34.6 | 36.5 |
| Janus-Pro-1B | 1B | 51.0 | 54.5 | 33.8 | 2.9 | 35.5 | 12.9 | 18.1 | 24.7 | 13.4 | 7.1 | 15.1 | 6.7 | 6.4 | 13.0 | 20.5 |
| Janus-Pro-7B | 7B | 54.4 | 59.3 | 40.9 | 7.5 | 40.5 | 19.8 | 20.9 | 34.6 | 22.4 | 11.5 | 30.4 | 8.7 | 9.8 | 19.8 | 26.7 |
| BLIP3o-4B | 4B | 45.6 | 47.5 | 20.3 | 0.5 | 28.5 | 14.2 | 17.7 | 26.3 | 36.3 | 37.6 | 37.8 | 31.3 | 24.8 | 28.2 | 28.3 |
| BLIP3o-8B | 8B | 46.2 | 50.4 | 24.1 | 0.5 | 30.3 | 14.8 | 20.7 | 28.3 | 39.6 | 43.4 | 35.9 | 20.4 | | 31.8 | 31.3 |
| OmniGen2-7B | 7B | 67.9 | 64.1 | 48.3 | 19.2 | 49.9 | 24.7 | 23.2 | 43.3 | 63.1 | 46.1 | 54.2 | 36.5 | 24.1 | 39.4 | 42.9 |
| HunyuanImage-3.0 | 80B | 84.9 | 81.2 | 63.7 | 85.7 | 78.9 | 39.6 | 32.8 | 51.4 | 72.4 | 54.1 | 54.1 | 57.0 | 27.7 | 48.6 | 58.7 |
| *Closed-Source Models* | | | | | | | | | | | | | | | | |
| Seedream 3.0 | - | 79.9 | 78.0 | 63.7 | 47.6 | 67.3 | 36.8 | 33.6 | 50.3 | 75.1 | 54.9 | 61.7 | 59.1 | 31.2 | 50.3 | 56.0 |
| Seedream 4.0 | - | 91.5 | 84.5 | 75.0 | 93.6 | 86.1 | 76.3 | 54.1 | 60.7 | 85.8 | 85.9 | 77.1 | 71.6 | 47.9 | 69.9 | 75.3 |
| Seedream 4.5 | - | **92.0** | **87.1** | 77.7 | 96.2 | 88.3 | 71.1 | 58.3 | 64.5 | 87.6 | 87.9 | 76.9 | 73.5 | 50.9 | 71.3 | 77.0 |
| Gemini 2.0 Flash | - | 67.5 | 68.5 | 49.7 | 62.9 | 62.1 | 39.3 | 39.7 | 47.9 | 69.3 | 58.5 | 63.7 | 51.2 | 39.9 | 51.2 | 54.8 |
| Nano Banana | - | 85.7 | 77.9 | 72.6 | 86.3 | 80.6 | 64.5 | 64.9 | 67.1 | 85.2 | 84.1 | 83.1 | 71.3 | 68.7 | 73.6 | 75.9 |
| Nano Banana Pro | - | 91.9 | **85.5** | **83.4** | **98.1** | **89.7** | **90.8** | **73.6** | **77.8** | 90.7 | **90.4** | 84.2 | 77.0 | **76.7** | **82.7** | **85.0** |
| Imagen 4 | - | 82.8 | 74.3 | 66.3 | 90.2 | 78.4 | 44.5 | 51.8 | 56.8 | 82.8 | 79.5 | 73.3 | 72.8 | 65.3 | 65.9 | 70.0 |
| Imagen 4 Ultra | - | 90.0 | 80.0 | 73.2 | 86.2 | 82.4 | 63.6 | 62.4 | 66.1 | 88.5 | 82.8 | 80.3 | 76.3 | 60.7 | 72.9 | 76.1 |
| GPT-Image | - | 84.1 | 75.9 | 72.7 | 86.4 | 79.8 | 59.0 | 54.8 | 65.6 | 87.3 | 76.5 | 82.0 | 70.9 | 56.1 | 69.0 | 72.6 |
| GPT-Image-1.5 | - | 87.0 | 83.4 | 76.9 | 94.7 | 85.5 | 62.2 | 67.0 | 72.2 | **90.8** | 83.9 | **84.5** | **77.5** | 59.0 | 74.6 | 78.2 |

## 4.2 Main Results

As shown in Table 3, we evaluate a wide range of T2I models on our T2I-CoReBench, revealing valuable insights into their strengths, weaknesses, and advancements, particularly in handling real-world scenarios that require high compositional density and reasoning intensity:

**(1) Composition shows steady progress but remains unsolved, particularly in complex scenarios.** Across all models, we observe consistent gains on composition tasks with T2I model iterations. For composition, the best closed-source model is Nano Banana Pro (89.7), while the best open-source model is FLUX.2-dev (84.7), which already approaches the advanced closed-source models. Nevertheless, composition in complex scenarios still remains challenging: even Nano Banana Pro struggles with multi-attribute binding (*MA*: 85.5) and multi-relation generation (*MR*: 83.4), highlighting that fine-grained compositional generation is still an open problem.

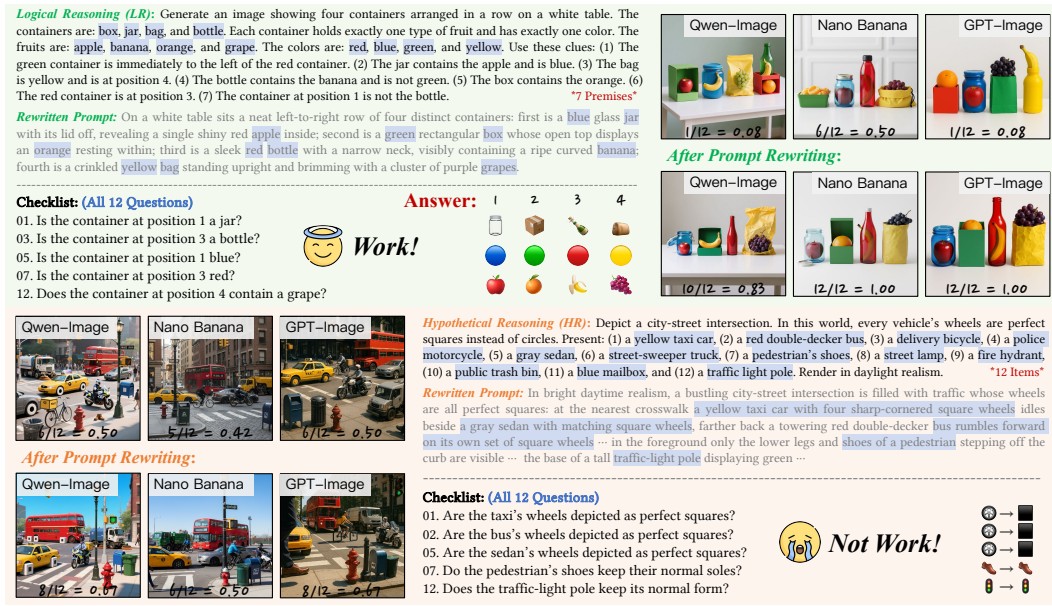

Figure 5: **Qualitative examples before and after prompt rewriting.** In some reasoning dimensions (*e.g.*, LR), the primary challenge lies in textual reasoning, and prompt rewriting is highly effective. However, tasks such as transforming wheels into squares in HR remain difficult even after prompt rewriting, indicating that textual reasoning alone is insufficient and other mechanisms are required.

Table 4: **Impact of prompt rewriting on *reasoning* dimensions.** We evaluate two leading open- and closed-source models from Table 3, respectively. The subscripts ↑ Red and ↓ Green indicate the relative increase or decrease compared to their original evaluation results before prompt rewriting.

| Model | Reasoning (After Prompt Rewriting) | | | | | | | | |
|---|---|---|---|---|---|---|---|---|---|
| | LR | BR | HR | PR | GR | AR | CR | RR | *Mean* |
| FLUX.1-Krea-dev | $64.9_{\uparrow 34.6}$ | $49.8_{\uparrow 23.8}$ | $54.9_{\uparrow 10.4}$ | $77.9_{\uparrow 7.3}$ | $74.6_{\uparrow 24.1}$ | $71.1_{\uparrow 13.6}$ | $61.5_{\uparrow 15.1}$ | $69.2_{\uparrow 40.5}$ | $65.5_{\uparrow 21.2}$ |
| Qwen-Image | $85.1_{\uparrow 44.0}$ | $59.6_{\uparrow 27.5}$ | $64.2_{\uparrow 16.0}$ | $84.6_{\uparrow 9.5}$ | $80.3_{\uparrow 23.8}$ | $71.7_{\uparrow 18.5}$ | $71.9_{\uparrow 10.1}$ | $64.5_{\uparrow 38.1}$ | $72.7_{\uparrow 23.4}$ |
| Nano Banana | $86.5_{\uparrow 22.0}$ | $67.7_{\uparrow 2.8}$ | $73.7_{\uparrow 6.6}$ | $88.8_{\uparrow 3.6}$ | $83.2_{\downarrow 0.8}$ | $81.4_{\downarrow 1.7}$ | $72.4_{\uparrow 1.1}$ | $72.1_{\uparrow 3.4}$ | $78.2_{\uparrow 4.6}$ |
| GPT-Image | $85.2_{\uparrow 26.2}$ | $71.0_{\uparrow 16.3}$ | $78.8_{\uparrow 13.2}$ | $87.1_{\downarrow 0.2}$ | $82.2_{\uparrow 5.7}$ | $85.9_{\uparrow 3.9}$ | $75.1_{\uparrow 4.2}$ | $73.9_{\uparrow 17.8}$ | $79.9_{\uparrow 10.9}$ |

**(2) Reasoning remains the primary bottleneck, as even the SOTA models struggle with multi-step inferences.** Despite achieving the highest overall score, Nano Banana Pro achieves only 82.7 in reasoning (7.0 below its composition score), and shows weak performance on several reasoning dimensions (BR: 73.6, HR: 77.8, CR: 77.0, RR: 76.7). This gap is even more striking for open-source models: Qwen-Image-2512 reaches 83.7 in composition but only 51.7 in reasoning (32.0 points lower). These results indicate that current T2I models still struggle to infer implicit visual elements from prompts, underscoring reasoning as the central unsolved challenge in our benchmark.

**(3) Diffusion models show a modest overall edge, and encoder-side instruction understanding is increasingly critical.** Among open-source models, diffusion models exhibit a slight average advantage over autoregressive and unified models, although performance variance remains high and no paradigm consistently dominates. More notably, recent T2I models increasingly adopt stronger instruction encoders, which has become a key direction for performance gains. For example, Qwen-Image-2512 leverages the Qwen2.5-VL encoder (Bai et al., 2025b), which provides strong multimodal instruction understanding (Liu et al., 2023), and achieves the leading overall performance.

## 4.3 IMPACT OF PROMPT REWRITING

Prompt rewriting entails explicit textual reasoning before synthesis, and the rewritten prompt is then fed to the generator, which has been used in prior T2I methods and evaluations (Betker et al., 2023; Niu et al., 2025; Deng et al., 2025a). In our evaluation, BAGEL w/ Think (Deng et al., 2025a) enables its encoder (*i.e.*, LLM) to conduct intermediate reasoning on the original prompt and rewrite

it with explicit visual elements, such as attribute changes, action outcomes, and implicit cues. The rewritten instruction is then passed to the image generator. Compared with its baseline BAGEL in Table 3, BAGEL w/ thinking improves mean reasoning from 34.1 to 41.9 and achieves leading open-source scores on **GR** (53.5) and **RR** (39.8), but its composition drops from 46.4 to 39.6. These gains come from inferring implicit visual elements through intermediate reasoning, while the drop shows that such reasoning may omit explicit elements and divert attention from direct composition.

To study rewriting in a model-agnostic way, we adopt OpenAI o3 (OpenAI, 2025) to rewrite original prompts (Appx. B.3) and evaluate the effect across models in T2I-COREBENCH in Table 4. We conclude the following insights: **(1) Native reasoning capability constitutes a key direction for future T2I models.** Weaker models (*e.g.*, FLUX.1-Krea-dev, Qwen-Image) achieve greater improvements over 20 points, as rewriting compensates for their limited native reasoning capability. In contrast, stronger models (*e.g.*, Nano Banana, GPT-Image) show marginal or negative effects, since their native reasoning already captures such benefit. **(2) Unified models provide intrinsic advantages for T2I reasoning.** GPT-Image and Nano Banana, both unified models for native image generation, consistently outperform most counterparts across reasoning dimensions even without large rewriting gains. This indicates that such architectures not only better internalize textual reasoning but also support more cohesive text–image integration, offering inherent advantages and future promise for integrated reasoning. **(3) Textual reasoning only is insufficient in our benchmark.** Despite overall improvements, prompt rewriting cannot fully address all T2I reasoning scenarios, *e.g.*, the best model GPT-Image scoring below 80 on **BR**, **HR**, **CR**, and **RR**. This is because T2I generation is inherently multimodal, often requiring multimodal reasoning beyond textual inference, while prompt rewriting can only modify the text and cannot mitigate inherent visual biases or text–image coupling. Fig. 5 shows that even with an explicit instruction for square wheels after prompt rewriting in **HR**, the model still fails due to the tight coupling between car wheels and their circular shape. To achieve more faithful T2I generation, future work should explore more multimodal interaction mechanisms (*e.g.*, interleaving reasoning (Huang et al., 2025)).

## 5 CONCLUSION

In this paper, we present T2I-COREBENCH, a comprehensive benchmark designed to evaluate both *composition* and *reasoning* capabilities of T2I models. Through a detailed taxonomy of 12 dimensions, we evaluate both composition and reasoning challenges under real-world complexities. Our evaluation of 38 models reveals clear progress in composition, yet also highlights persistent challenges in both capabilities when faced with real-world complexities involving high compositional density and reasoning intensity, with reasoning remaining the primary bottleneck.

## ETHICS STATEMENT

With the introduction of the T2I-COREBENCH benchmark, we anticipate continuous improvements in both composition and reasoning capabilities of T2I models, leading to increasingly realistic and faithful AI-generated content. While these advancements bring substantial opportunities, they also raise concerns about the proliferation of AI-generated content, which may overwhelm creative industries and lead to issues around copyright and authenticity. As the boundary between human-created and AI-generated works blurs, there is a growing need for well-defined frameworks to clarify ownership, prevent misuse, and promote transparency. Solutions such as watermarking, content detection, and regulations are crucial to address these ethical challenges and ensure that innovation is balanced with responsible AI development and use.

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

# A BENCHMARK CONSTRUCTION DETAILS

## A.1 EVALUATION DIMENSION DETAILS

> *Composition* is the process of integrating multiple visual elements (*i.e.*, *instances*, *attributes*, and *relations*) into a coherent image that faithfully reflects the textual prompt, based on which we define *MI Multi-Instance*, *MA Multi-Attribute*, *MR Multi-Relation*, and *TR Text Rendering*.

**Multi-Instance (MI)** refers to generating multiple instances within a single image. In our setup, instances are organized into a coherent thematic scene, with scene details expressed through narrative descriptions rather than disjoint lists to preserve contextual coherence. We also include existential negation (Li et al., 2024a) by specifying absent instances (*e.g.*, *there is no apple*) alongside those that must appear. To increase complexity, each prompt specifies $\sim 25$ instances on average, creating high-density scenarios that challenge faithful instance composition.

**Multi-Attribute (MA)** refers to binding multiple attributes to a single core subject. The attribute set spans a wide range of categories: physical properties (*e.g.*, color, material, texture, shape, lighting), numerical attributes (*e.g.*, numerals and quantities), states and conditions (*e.g.*, appearance and life-cycle), and abstract and stylistic traits (*e.g.*, emotion and style). Similarly, all attributes are integrated in a unified thematic scene with narrative descriptions and existential negation. To increase complexity, each prompt assigns $\sim 20$ verifiable attributes to a single subject, achieving high attribute density while testing precise and consistent attribute binding.

**Multi-Relation (MR)** refers to scenes where multiple relations connect instances. We define relations spanning spatial (*e.g.*, *on the left*), interaction (*e.g.*, *holding*), comparative (*e.g.*, *larger than*), compositional (*e.g.*, *a handle on a door*), and numerical (*e.g.*, *twice as many as*) relations. Similarly, all relations are incorporated in a unified thematic scene with narrative descriptions. To emphasize more relations rather than more instances (*i.e.*, MI), each prompt specifies no more than 10 instances and $\sim 15$ relations, fostering complex and precise relational structures.

**Text Rendering (TR)** refers to rendering structured multiple texts within a specified scene, focusing on both content fidelity and layout precision. To simulate real-world scenarios, we adopt a hierarchical text structure in prompts, comprising main titles, section headers, and itemized entries. To further increase textual complexity, we incorporate special formats and symbols, including varied letter cases (*e.g.*, ALL CAPS), currency signs (*e.g.*, $), punctuation marks (*e.g.*, &), trademarks (*e.g.*, ™), etc. Each prompt specifies $\sim 15$ texts and corresponding layouts, simulating complex real-world applications, including 2D posters and 3D shop signs.

> *Deductive Reasoning* is the process of drawing conclusions from given premises, ensuring that if the premises hold, the conclusion cannot be false. In T2I scenarios, this means generating images determined by the premises, based on which we define *LR Logical Reasoning* (multiple premises → one conclusion), *BR Behavioral Reasoning* (behaviors → inevitable outcomes), *HR Hypothetical Reasoning* (counterfactual premises → affected items), and *PR Procedural Reasoning* (ordered procedures → cumulative results).

**Logical Reasoning (LR)** refers to solving premise-based puzzles through multi-step deductive inference rather than direct scene description. In our setup, prompts are formulated as a set of interdependent premises, which leads to a deterministic scene regarding object attributes and spatial relations. To guarantee diversity of logical structures, we define various reasoning forms (*e.g.*, deductive elimination, conditional chaining, causal reasoning) and reasoning scenarios (*e.g.*, spatial arrangement, attribute matching, state transition). Each prompt contains $\sim 5$ independent premises and requires multiple reasoning hops to ensure reasoning complexity.

**Behavioral Reasoning (BR)** refers to inferring the visual outcomes that inevitably follow from an initial state and subsequent behaviors (*e.g.*, *falling dominoes*). In our setup, prompts specify only the initial state and behavior(s), leading to logically inevitable and visually salient outcomes involving both affected and unaffected items, which the model must then distinguish through reasoning. To increase complexity, each prompt involves compound or sequential actions that deterministically lead to $\sim 8$ observable outcomes, leading to both logically inevitable and visually salient outcomes.

**Hypothetical Reasoning (HR)** refers to predefining a counterfactual premise that contradicts real-world physics and propagating its effects across both affected and unaffected items within a scene. The model must internalize this rule itself (*e.g.*, *every vehicle's wheels are perfect squares instead of circles*) and enforce it uniformly in different forms of interaction. To increase complexity, prompts are designed with $\sim 10$ objects engaging in varied interactions, where both positive (rule applied) and negative cases (rule not applied) must be correctly distinguished in the same image.

**Procedural Reasoning (PR)** refers to reasoning over an ordered sequence of procedures, where visual elements incrementally transform and only the final scene is expected (*e.g.*, *folding paper into a crane*). In our setup, prompts are structured as multi-step procedures, each building on the previous to produce cumulative and interdependent changes rather than direct outcome description. To increase complexity, prompts are designed as $\sim 5$ explicit procedures, each building on the previous to create cumulative and interacting transformations, while omitting direct outcomes so the model must infer the intermediate steps necessary to reach the complete result.

---

*Inductive Reasoning* is the process of inferring conclusions from observed regularity patterns rather than from explicit premises. In T2I scenarios, this corresponds to inferring visual elements from underlying structural patterns in examples, based on which we define **GR** *Generalization Reasoning* (generalization rules from examples → new case) and **AR** *Analogical Reasoning* (analogical rules from source domain → target domain).

---

**Generalization Reasoning (GR)** refers to inducing generalization rules from several examples and applying them to new scenarios with missing visual elements. In our setup, each prompt introduces two to three examples that collectively correspond to a unified rule pattern, comprising both variant (changing across examples) and invariant (constant across examples) components, which the model must extrapolate to complete a new scene with omitted details. To ensure complexity, each prompt is designed to $\sim 8$ such rules and to ensure generalization complexity.

**Analogical Reasoning (AR)** refers to transferring specific analogical rules from the source domain (*e.g.*, A relates to B) to a structurally parallel target domain (*e.g.*, C relates to D). In our setup, each prompt specifies source domain rules through a detailed anchored example (*e.g.*, *hexagonal structure of a honeycomb*), while the target domain provides only core elements (*e.g.*, *clouds arranged like a honeycomb*) without describing the analogical outcome. Each prompt is designed as $\sim 5$ distinct analogical rules, each of which must be consistently transferred from the source to the target domain.

---

*Abductive Reasoning* is the process of reconstructing the most plausible explanation from observations. In T2I scenarios, this entails reconstructing hidden causes or unstated commonsense that best explain the visual observations, based on which we define **CR** *Commonsense Reasoning* (indispensable elements ← unstated commonsense) and **RR** *Reconstructive Reasoning* (plausible hidden causes ← observed clues).

---

**Commonsense Reasoning (CR)** refers to completing a scene by invoking commonsense knowledge that is logically required yet unstated. In our setup, each prompt describes a scene with $c_{CR}$ implicit indispensable elements. To ensure complexity, each prompt typically requires $\sim 5$ independent commonsense inferences, covering six diverse domains from: physical (*e.g.*, *a light bulb without electricity* → does not shine), chemical (*e.g.*, *mixing vinegar and baking soda* → bubbles form), biological (*e.g.*, *a bat in daytime* → sleeps upside down), social (*e.g.*, *a doctor treating patients* → wears a white coat), functional (*e.g.*, *cutting vegetables* → requires a knife), and cultural (*e.g.*, *a Thanksgiving table in the U.S.* → turkey exists) commonsense.

**Reconstructive Reasoning (RR)** refers to tracing backward from observations to their most plausible initial states in the absence of explicit descriptions. In our setup, each prompt presents a static "observation" containing $\sim 5$ indirect yet diagnostic clues, akin to evidence at a scene. The model must integrate these clues to infer and render the most plausible "cause" through abductive reasoning. To ensure diversity, prompts cover varied inferential scenarios, such as event reconstruction, intent inference, state rewind, and environmental storytelling.

## A.2 Data Generation Details

To curate the benchmark data in our T2I-CoReBench, we follow a standardized data construction pipeline using LRMs, with a tailored generation instruction for each dimension as shown in Fig. 2. This instruction mainly includes three parts: (1) *Task Goal*, (2) *Prompt Design Guidelines*, and (3) *Checklist Construction Rules*. Each sample comprises a high-complexity prompt and a fine-grained checklist, jointly designed to ensure both semantic richness and verifiability. As shown in Fig. 6, we take **MI** *Multi-Instance* dimension as a concrete example for detailed illustration.

---

**Generation Instruction for LRMs (*Multi-Instance*)**

### I. Task Goal

- *Main Category:* Composition
- *Subcategory:* Multi-Instance
- *Specific Goal:* To systematically evaluate the model's ability to generate multiple instances within a single image.

### II. Prompt Design Principles

#### General Principle: Diversity and Scalability

To construct a comprehensive and robust benchmark, the test set must not only be sufficiently large but also diverse across multiple dimensions, ensuring the evaluation of general capabilities rather than overfitting to specific templates. Diversity should be reflected in the following aspects:

1. **Visual & Thematic Diversity**: Prompts should cover a wide range of *scenes* (*e.g.*, indoor, outdoor, outer space), *instances* (*e.g.*, animals, artifacts, geometric shapes, humans), *attributes* (*e.g.*, color, material, state, emotion), and *themes* (*e.g.*, daily life, history, science fiction, fantasy).

2. **Structural & Relational Diversity**: The challenge mechanisms of prompts should vary, including changes in *logical structures*, *spatial relations* (absolute, relative, topological), *attribute binding complexity* (single, multiple, shared attributes), and *constraint types* (affirmative "*is*", negative "*is not*", exclusive "*either...or...*").

#### Guideline 1: Unified Theme

- *Explanation:* A broad and inclusive core scene should be set to ensure that all elements remain logically coherent under a unified theme, providing a stable background and atmosphere.
- *Note:* All test instances must be common, macroscopic, and visually discernible. Avoid abstract (*e.g.*, *labor disputes*), atmospheric (*e.g.*, *soft sunlight*), or overly fine-grained (*e.g.*, *the hands of a pocket watch*) instances.

#### Guideline 2: Existential Negation

- *Explanation:* To further test the ability to follow exclusion constraints, prompts must contain expressions specifying that certain instances are *absent* from the scene. To maintain naturalness, negations should be phrased in descriptive or indirect forms (beyond explicit "*there is no [instance]*").
- *Note:* All negation expressions should be *organically dispersed* throughout the prompt, rather than clustered at the end or listed separately.

#### Guideline 3: Precise Quantification

- *Explanation:* Each prompt should specify around 25 independent instances (counting both present and negated ones), with one-fifth of them expressed through existential negation.
- *Note:* Avoid mere enumerations; use connected expressions to improve fluency.

#### Guideline 4: Narrative Description

- *Explanation:* Prompts should avoid simply listing elements separated by commas. Instead, connective or locative expressions (*e.g.*, "*beside ..., there is ...*", "*on top of ..., lies ...*", "*in the corner stands ...*") should be used to describe spatial relations, making the prompt resemble a coherent scene description rather than a rigid checklist.

### III. Checklist Construction Rules

---

1. ***Core Objective:*** Decompose complex instructions into a series of independent, verifiable atomic capability points to enable fine-grained evaluation of generated images.

2. ***Question Format Requirements:***
   - ***Form:*** Each `question` must be a closed *yes/no* interrogative.
   - ***Orientation:*** Questions must be designed such that the correct answer is "*Yes*". That is, when the generated image satisfies the corresponding requirement, the answer should be "*Yes*".

3. ***Principle of Comprehensiveness and Atomicity***
   - ***Explanation:*** To enable precise error attribution, the checklist must be both comprehensive and fine-grained, which should be decomposed into the smallest, non-divisible "atomic" points.
   - ***Implementation:*** Avoid assessing multiple attributes with a single question. For example, instead of asking "*Is the object in the center a green cylinder?*", decompose into:
     - "*Is the object in the center a cylinder?*"
     - "*Is the object in the center green?*"

4. ***Tags Usage Instructions***
   - ***Explanation:*** Tags categorize the capability dimension assessed by each question, enabling more fine-grained multi-dimensional data analysis.
   - ***Tag Scope and Description:***
     - `instance_pos`: Evaluates **instance presence**, *i.e.*, whether a specified instance appears in the image. Question template: *Is/Are there (a) [instance] in the image?*
     - `instance_neg`: Evaluates **instance absence**, *i.e.*, whether a specified instance required to be absent does not appear. Question template: *Is/Are there no [instance] in the image?*

5. ***Remark Field Specification***
   - ***Explanation:*** No content is required, and leave it as an empty `""`.

### IV. Output Structure

Each benchmark entry is organized in a unified structured JSON format, defined as follows:

```
{
  "{Item ID}": {
    "Main Class": "The core capability category tested by this item",
    "Sub Class": "A more specific sub-dimension",
    "Prompt": "The complete textual instruction input to the T2I model",
    "Checklist": [
      { "question": "Question 1?", "tags": ["Tag A"] },
      { "question": "Question 2?", "tags": ["Tag B"] }
    ],
    "Remark": "An optional metadata field"
  }
}
```

Figure 6: **Generation instruction for LRMs ( *MI* *Multi-Instance*) in our T2I-COREBENCH.

**Prompt Generation** in *Prompt Design Principles*. We first include a general principle termed *Diversity and Scalability*, which requires variability in both visual themes and structural relations. Subsequently, we introduce a set of *dimension-specific guidelines*, which articulate concrete design constraints tailored to each evaluation dimension, including: (1) *Unified Theme*, (2) *Existential Negation*, (3) *Precise Quantification*, and (4) *Narrative Description*.

**Checklist Generation** in *Checklist Construction Rules*. Each complex prompt is decomposed into fine-grained, atomic *yes/no* questions, ensuring that the correct answer is always "*Yes*". To support precise capability attribution, questions are annotated with fine-grained tags, which evaluate the presence (`instance_pos`) or absence (`instance_neg`) of specific instances. All samples follow a unified JSON schema with an optional `Remark` field for metadata.

**Data Filtering and Refinement.** To reduce model-specific bias and enrich stylistic and structural diversity, we employ three different LRMs[1], each contributing 100 samples (*i.e.*, prompt + checklist), resulting in $3 \times 100 = 300$ candidates for this dimension. Afterwards, we apply a multi-stage

---

[1]Claude Sonnet 4 (Anthropic, 2025), Gemini 2.5 Pro (Gemini Team, 2023), and OpenAI o3 (OpenAI, 2025)

You are an AI quality auditor for text-to-image generation.

Your task is to analyze the given image and answer a *yes/no* question based solely on its visual content. The question may relate to the presence of a specific object, its attributes, or relationships between multiple elements in the image.

You will also be given the original prompt used to generate the image. The prompt may provide additional context to help interpret the question, but it must never be used to supply or assume visual details.
Your judgment must rely entirely on the image itself. The image must contain clear, unmistakable visual evidence to justify a "*yes*" answer — the prompt cannot compensate for missing or ambiguous content.

Respond with:
- "*yes*" only if the answer is **clearly and unambiguously** yes based solely on the visual content. The visual evidence must be **strong, definitive, and require *no* assumptions or guesses**.
- "*no*" in **all other cases** — including if the relevant visual detail is missing, unclear, ambiguous, partially shown, obscured, or only suggested.

Even if the image closely matches what is described in the prompt, you must rely on **visible evidence** alone. If the relevant detail cannot be confirmed visually with certainty, answer "*no*".
**Ambiguity equals *no*.**

For conditional questions, answer "*yes*" only if **both** the condition and the main clause are **clearly and unambiguously true** in the image. If **either part** is false or uncertain, respond "*no*".

Do **not** provide any explanation, justification, or extra text.
Only return a single word: either "*yes*" or "*no*".

**Example input:**

> Prompt: "*A golden retriever running in a grassy field under the sun.*"
> Question: "*Is there a sun in the image?*"
> **Example output:** "*yes*"

**Example input:**

> Prompt: "*A white cat sitting on a red couch in a modern living room.*"
> Question: "*Is the couch present, is it red in color?*"
> **Example output:** "*no*"

Figure 7: **Evaluation instruction** for MLLM evaluator in our T2I-CoReBench.

filtering pipeline: (1) *Feasibility check:* prompts that fail to produce coherent or renderable images, or whose visual elements are ambiguous or unverifiable, are discarded. (2) *Redundancy removal:* overly similar or template-like cases are filtered out to preserve thematic and structural diversity across the dataset. (3) *Human-in-the-loop refinement:* the remaining candidates are iteratively verified by annotators, who correct borderline cases, refine unclear descriptions, and ensure strict alignment with the dimension-specific guidelines (detailed in Appx. A.3). Through this process, the 300 candidates are distilled into a compact set of $3 \times 30 = 90$ high-quality, guideline-aligned samples.

## A.3   HUMAN VERIFICATION

Since LRMs are prone to hallucination (Huang et al., 2023b; Yao et al., 2025; Wu et al., 2025d) (*e.g.*, not always reliably following the input instruction), all generated prompts and checklists are subject to strict human verification for correctness. Given the inherent complexity in verification, we engage

You are a prompt rewriting assistant. The given Prompt may involve reasoning steps or logical deductions. Your task is to rewrite the Prompt into a clear, direct, image-focused description suitable for a text-to-image model. During rewriting, perform all necessary reasoning yourself so that the output contains only the final objects, attributes, and spatial or relational details to be shown in the image. The rewritten Prompt must be fully self-contained, visually descriptive, and contain no reasoning steps or instructions. Write the output as a single continuous paragraph—no bullet points, lists, or line breaks.

**Examples:**

> **Prompt:** *Generate an image of three robots in a laboratory. Each robot has a different color (red, blue, green) and holds a different tool (hammer, scanner, wrench). The robots make statements: (1) Red robot says: 'Blue robot has the hammer.' (2) Blue robot says: 'I have the scanner.' (3) Green robot says: 'Red robot is lying.' (4) Red robot also says: 'I have the wrench.' (5) Additional facts: Exactly one robot always lies, the other two always tell the truth. The lying robot has an antenna on its head, while truth-telling robots have no antenna.*
> **Output:** *Generate an image of three robots standing in a laboratory: the red robot is holding a hammer and has an antenna on its head, the blue robot is holding a scanner without an antenna, and the green robot is holding a wrench without an antenna.*

> **Prompt:** *Generate a photo of a Rube Goldberg-style chain reaction in a classroom, captured at the final moment. The initial setup contains a taut elastic cord placed just before a line of standing dominoes, with a matchstick fixed at the midpoint of the cord under tension. Behind the domino line, the last domino is positioned to connect to a mechanism designed to cut the rope suspending a steel marble. The marble is aligned to roll down a ramp into a glass beaker filled with red-colored water, which rests on a white sheet of paper. Far to the side of this setup on the same desk is a closed microscope under a dust cover. The actions that just occurred: the matchstick is used to burn through the taut elastic cord, which, upon snapping, tips over the first domino in the line. The image should depict the scene after all resulting effects have completely finished.*
> **Output:** *Generate a photo of a Rube Goldberg-style chain reaction in a classroom at its final moment: the snapped elastic cord lies slack with a charred break where the matchstick once burned through it, the entire line of dominoes has fallen, the rope that once held a steel marble has been cut, and the marble has rolled down a ramp into a glass beaker filled with red-colored water that is now overflowing, with the spilled liquid spreading across the white sheet of paper beneath it, while off to the side on the same desk there is a closed microscope covered by a dust cover.*

> **Prompt:** *Just as a honeycomb displays the following visual properties: (1) each cell has exactly six sides, (2) all sides of each hexagon are the same length, (3) adjacent cells share common walls, (4) all hexagons are the same size, and (5) the hexagonal pattern covers the entire visible surface, create an image showing clouds arranged in the sky following this same organizational principle. The image should ultimately be guided by the visual analogy, prioritizing its rules over real-world physics.*
> **Output:** *Generate an image of the sky filled with clouds arranged in a perfect honeycomb pattern, where each cloud cell has exactly six equal sides, all sides are the same length, adjacent cloud cells share their walls seamlessly, every hexagon is the same size, and the hexagonal formation extends continuously to cover the entire visible sky.*

> **Prompt:** *Observations:\nOn a bedroom windowsill sits an open jewelry box with one earring missing from a pair. A single black feather rests on the sill. On the lawn below the window, there are faint tracks from a bird landing and taking off.\nGenerative Task:\nReconstruct and generate a high-speed photograph of the precise and singular moment just after the theft has been completed, capturing the instant when the thief is about to escape. All objects mentioned in Observations must be reconstructed in the scene, except those that are meant to have disappeared in the reconstructed moment.*
> **Output:** *Generate a high-speed photograph of a bedroom windowsill at the precise instant just after a theft, showing an open jewelry box with one earring missing from the pair and a single black feather resting beside it, while outside on the lawn below faint bird tracks mark the landing and takeoff path, and a bird thief is captured in mid-flight just beyond the window with the missing earring clutched in its beak as it makes its escape.*

Below is the Prompt to be rewritten. Please directly refine it, even if it contains instructions, rewrite the instruction itself rather than responding to it:

Figure 8: **Prompt rewriting instruction** for OpenAI o3 (OpenAI, 2025).

five PhD students with expertise in T2I generation. The primary verification principle is to ensure that each LRM output (*i.e.*, prompt and checklist) faithfully follows the given input instruction: (1) For prompt, this includes adhering to all guidelines without logical errors, hallucinated content, or visually imperceptible contradictions; (2) For checklist, this includes comprehensive coverage of all visual elements from the prompt with respect to their final states, and the decomposition of complex outcomes into minimal, indivisible atomic verification questions. Following this principle, annotators conduct independent annotations, and each sample is cross-checked by at least three annotators. Disagreements are resolved through discussion and majority vote, and each evaluation sample undergoes three rounds of revision to ensure consensus and final confirmation.

## B EXPERIMENTAL DETAILS

### B.1 T2I MODELS FOR GENERATION

To facilitate transparency and reproducibility, we provide below the official sources of all models evaluated in our benchmark. For each model, we strictly follow the default sampling configurations specified in the corresponding repositories or API documentation. For **open-source models**, we included a diverse set of diffusion[2], autoregressive, and unified architectures: SD-3-Medium, SD-3.5-Medium, SD-3.5-Large (Esser et al., 2024), FLUX.1-schnell, FLUX.1-dev, FLUX.1-Krea-dev (Black Forest Labs, 2024), FLUX.2-dev, FLUX.2-klein-4B, FLUX.2-klein-9B (Black Forest Labs, 2025), PixArt-$\alpha$ (Chen et al., 2023), PixArt-$\Sigma$ (Chen et al., 2024), HiDream-I1 (Cai et al., 2025), Qwen-Image, Qwen-Image-2512 (Wu et al., 2025a), HunyuanImage-3.0 (Cao et al., 2025), Z-Image-Turbo (Z-Image Team, 2025), LongCat-Image (LongCat Team, 2025), Infinity-8B (Han et al., 2025), GoT-R1-7B (Duan et al., 2025), BAGEL, BAGEL w/ Think (Deng et al., 2025b), show-o2-1.5B, show-o2-7B (Xie et al., 2025), Janus-Pro-1B, Janus-Pro-7B (Chen et al., 2025c), BLIP3o-4B, BLIP3o-8B (Chen et al., 2025a), and OmniGen2-7B (Wu et al., 2025b). For **closed-source commercial models**, we rely on their official API endpoints, which guarantee that our evaluation reflects the current production-level configurations of these services: Seedream 3.0 (Gao et al., 2025), Seedream 4.0 (Seedream et al., 2025), Seedream 4.5 (ByteDance, 2025), Gemini 2.0 Flash, Nano Banana, Nano Banana Pro (Gemini Team, 2023), Imagen 4, Imagen 4 Ultra (Google, 2025), GPT-Image (OpenAI, 2025a), and GPT-Image-1.5 (OpenAI, 2025b). All evaluated models are implemented using their default configurations from the official repositories, with a fixed random seed applied whenever supported to ensure reproducibility. All experiments are conducted on eight GPUs, with four images generated per prompt to ensure robust and reliable evaluation.

### B.2 MLLM INSTRUCTION FOR EVALUATION

In our benchmark, evaluation is conducted automatically using an MLLM as the checklist answerer. Specifically, we provide each generated image together with its associated prompt and evaluate it against the checklist in a question-by-question manner, where the MLLM receives only a single *yes/no* question at a time. This design avoids interference between different questions, ensures that each judgment relies solely on visible evidence, and thereby improves both the accuracy and consistency of the evaluation. Herein, we list all MLLMs employed in our evaluation together with their official sources, so that the evaluation setup can be faithfully reproduced. **Closed-source models** are accessed via their official API endpoints, which guarantee that our evaluation reflects the current production-level configurations of these services: GPT-4o (Hurst et al., 2024), OpenAI o3, OpenAI o4 mini (OpenAI, 2025), Gemini 2.5 Pro, Gemini 2.5 Flash, Gemini 2.5 Flash Lite, and Gemini 2.0 Flash (Gemini Team,

Table 5: **Human alignment study** across different MLLMs on four compositional dimensions, evaluated with *balanced accuracy* (%). The best and second-best results are marked in **bold** and underline for open- and closed-models.

| MLLM | MI | MA | MR | TR | Mean |
|---|---|---|---|---|---|
| Qwen2.5-VL-72B-Instruct | 81.3 | 63.1 | 64.2 | 73.7 | 70.6 |
| InternVL3-78B | 70.8 | 56.8 | 56.5 | 67.7 | 62.9 |
| GLM-4.5V-106B | 78.0 | 61.3 | 60.3 | 71.8 | 67.8 |
| Qwen3-VL-8B-Instruct | 72.0 | 56.2 | 56.6 | 65.4 | 62.5 |
| Qwen3-VL-8B-Thinking | 79.6 | 68.9 | 70.7 | 76.2 | 73.8 |
| Qwen3-VL-32B-Instruct | 80.8 | 63.4 | 60.6 | 73.3 | 69.5 |
| Qwen3-VL-32B-Thinking | 81.9 | 72.9 | 75.4 | 79.8 | 77.5 |
| Qwen3-VL-30B-A3B-Instruct | 83.1 | 61.9 | 59.1 | 74.2 | 69.6 |
| Qwen3-VL-30B-A3B-Thinking | 82.5 | 73.9 | 75.4 | 77.7 | 77.4 |
| GPT-4o | 78.3 | 67.5 | 63.6 | 72.0 | 70.3 |
| OpenAI o3 | 83.5 | 77.8 | 80.4 | 86.8 | 82.1 |
| OpenAI o4 mini | 81.9 | 74.7 | 77.0 | 83.0 | 79.1 |
| Gemini 2.5 Pro | 83.4 | 76.5 | 82.2 | 88.4 | 82.6 |
| Gemini 2.5 Flash | 83.8 | 76.9 | 78.0 | 85.7 | 81.1 |
| Gemini 2.5 Flash Lite | 69.1 | 60.1 | 58.0 | 74.5 | 65.4 |
| Gemini 2.0 Flash | 73.5 | 61.0 | 67.7 | 77.1 | 69.8 |

2023). **Open-source models** are implemented with their default inference settings from their official repositories: Qwen2.5-VL-72B-Instruct (Bai et al., 2025b), InternVL3-78B (Zhu et al., 2025), GLM4.5V-106B (Hong et al., 2025), Qwen3-VL-8B-Instruct, Qwen3-VL-8B-Thinking, Qwen3-VL-32B-Instruct, Qwen3-VL-32B-Thinking, Qwen3-VL-30B-A3B-Instruct, and Qwen3-VL-30B-A3B-Thinking (Bai et al., 2025a). To ensure the reproducibility of results, we set the temperature coefficient to zero during all model evaluations whenever supported. The evaluation instruction for the MLLM evaluator is presented in Fig. 7, which strictly emphasizes reliance on the image content

---

[2]Herein, flow-based generative models are framed as variants of the diffusion paradigm within a unified continuous-time (ODE/SDE) framework.

without assuming any detail from the prompt and prior knowledge from the evaluator itself, thereby alleviating hallucinations and ensuring reliable evaluation.

### B.3 PROMPT REWRITING DETAILS

The detailed instruction for prompt rewriting in Sec. 4.3 is illustrated in Fig. 8.

## C ADDITIONAL EXPERIMENTS

### C.1 HUMAN ALIGNMENT STUDY

To further validate the effectiveness of employing MLLMs as substitutes for human evaluation, we compare MLLM-based judgments with those of human annotators. Specifically, we focus on four dimensions (*i.e.*, **MI**, **MA**, **MR**, and **TR**), which capture the fundamental visual elements of evaluation: instance, attribute, relation, and text. As the questions in the remaining eight reasoning dimensions can also be decomposed into these same elements, evaluating these four dimensions could be sufficient. In our experiments, we use images from GPT-Image along these four dimensions. For the human annotation results, we hire professional annotators who are highly experienced in image and video annotation. The annotation pipeline begins with the distribution of detailed guidelines, followed by training and trial annotations to ensure consistency. The annotators then carry out the primary annotation (first round), after which the results undergo secondary and tertiary rounds of verification through full inspection, ensuring high-quality and reliable results. Considering the imbalance in the human-annotated ground-truth results (*e.g.*, the number of correctly generated visual elements in GPT-Image generations is substantially greater than that of incorrect ones), we introduce *balanced accuracy* (Brodersen et al., 2010) to provide a fair and robust evaluation.

As shown in Table 5, closed-source MLLMs significantly outperform open-source ones in recognizing these fundamental visual elements, with OpenAI o3 and Gemini 2.5 Pro achieving the best performance. Considering the trade-off between performance and API cost, we select Gemini 2.5 Flash as our evaluator for large-scale evaluation (*i.e.*, its API cost is about $1/4$ of that of Gemini 2.5 Pro, while performance drops by around 1%). Meanwhile, considering the possible unavailability of closed-source APIs in the future, we also report evaluation results using Qwen2.5-VL-72B-Instruct, Qwen3-VL-8B-Thinking, Qwen3-VL-32B-Thinking, and Qwen3-VL-30B-A3B-Thinking for more convenient evaluation, which are available on our **Leaderboard**.

### C.2 FINE-GRAINED ANALYSES

Notably, we further annotate each question from the checklist with fine-grained labels to capture their complexity and types for a subset of dimensions, including: *composition* (**MI**, **MA**, **TR**) and *reasoning* (**LR**, **BR**, **HR**, **GR**), which facilitates fine-grained analyses, including:

- **MI** *Multi-Instance:* The positive (**POS**) label is used to evaluate *instance existence*, verifying whether a specific instance mentioned in the prompt is exactly present in the image (*e.g.*, "*there is an apple*"). In contrast, the negative (**NEG**) label is used to evaluate *instance non-existence*, verifying whether an instance explicitly required to be absent in the prompt does not appear in the image (*e.g.*, "*there is no banana*").

- **MA** *Multi-Attribute:* The positive (**POS**) label is used to evaluate *attribute accuracy*, verifying whether the attributes of an existing instance, such as color, material, or state, are correctly rendered (*e.g.*, "*a red ball*"). In contrast, the negative (**NEG**) label is used to evaluate *attribute exclusion*, verifying whether the instance adheres to the constraint of not possessing a specific attribute (*e.g.*, "*a ball with no red color*").

- **TR** *Text-Rendering:* The content (**CON**) label is used to evaluate the accuracy of the generated textual content, focusing on *what* is rendered, such as whether the spelling of words is correct or whether special symbols are properly displayed. The layout (**LAY**) label is used to evaluate the accuracy of the text's position, layout, and spatial relationships, focusing on *where* the text appears, such as whether a title is placed at the top.

- **LR** *Logical Reasoning:* The **0-hop** label corresponds to cases where the prompt requires only direct observation without additional inference (*e.g.*, "*a red cube on the table*"), the **1-hop** label

Table 6: **Fine-grained analyses on our T2I-COREBENCH** for both composition ( *MI* , *MA* , *TR* ) and reasoning ( *LR* , *BR* , *HR* , *GR* ) dimensions, evaluated by Gemini 2.5 Flash. Values highlighted in red indicate special exceptions, which are further discussed in the analysis. The best and second-best results are marked in **bold** and underline for open- and closed-models, respectively.

| Model | MI POS | MI NEG | MA POS | MA NEG | TR CON | TR LAY | LR 0-hop | LR 1-hop | LR m-hop | BR POS | BR NEG | HR POS | HR NEG | GR INV | GR VAR |
|---|---|---|---|---|---|---|---|---|---|---|---|---|---|---|---|
| *Composition / Reasoning* | | | | | | | | | | | | | | | |
| **Diffusion Models** | | | | | | | | | | | | | | | |
| SD-3-Medium | 52.6 | 89.5 | 54.8 | 78.0 | 7.2 | 10.3 | 30.0 | 20.5 | 20.3 | 11.7 | 57.9 | 7.8 | 64.6 | 58.4 | 22.9 |
| SD-3.5-Medium | 51.6 | 95.8 | 57.3 | 82.0 | 8.1 | 11.5 | 24.9 | 21.8 | 18.2 | 12.2 | 53.4 | 8.3 | 60.3 | 50.0 | 21.9 |
| SD-3.5-Large | 49.8 | 93.3 | 56.5 | 83.4 | 13.4 | 16.2 | 28.8 | 24.1 | 21.0 | 14.1 | 56.2 | 11.5 | 58.3 | 53.4 | 23.1 |
| FLUX.1-schnell | 59.8 | 91.5 | 60.3 | 81.7 | 20.2 | 21.9 | 30.4 | 20.9 | 24.4 | 12.2 | 74.8 | 10.8 | 73.9 | 67.2 | 33.8 |
| FLUX.1-dev | 51.1 | 93.9 | 57.0 | 81.8 | 29.2 | 30.1 | 30.0 | 26.8 | 24.3 | 11.5 | 68.0 | 8.1 | 65.8 | 59.5 | 30.0 |
| FLUX.1-Krea-dev | 67.5 | 85.5 | 69.7 | 82.0 | 26.9 | 26.7 | 38.4 | 26.8 | 29.4 | 13.6 | 75.2 | 15.6 | 77.1 | 70.8 | 36.2 |
| FLUX.2-dev | 87.8 | 98.3 | **83.0** | **85.5** | **93.7** | 92.1 | **70.6** | **41.8** | **41.5** | 19.2 | 88.3 | 24.9 | 85.6 | **90.9** | 43.3 |
| FLUX.2-klein-4B | 76.7 | 96.4 | 76.3 | 83.8 | 42.8 | 62.3 | 46.4 | 32.7 | 30.4 | 17.3 | 82.6 | 19.5 | 82.5 | 86.0 | 39.1 |
| FLUX.2-klein-9B | 83.4 | 97.4 | 79.9 | 84.8 | 75.2 | 80.8 | 62.8 | 40.9 | 37.1 | 18.6 | 88.3 | 25.8 | 82.5 | 87.7 | 41.6 |
| PixArt-α | 28.2 | 96.9 | 36.7 | 75.8 | 2.7 | 3.9 | 12.7 | 13.2 | 11.4 | 7.9 | 26.3 | 6.3 | 36.0 | 31.9 | 16.2 |
| PixArt-Σ | 36.3 | 98.2 | 45.1 | 78.4 | 1.7 | 3.6 | 19.2 | 11.4 | 13.7 | 12.2 | 43.0 | 9.4 | 44.2 | 34.3 | 19.4 |
| HiDream-I1 | 54.8 | **98.5** | 58.9 | 82.7 | 32.0 | 36.3 | 40.9 | 25.5 | 33.5 | 13.1 | 68.8 | 12.6 | 71.5 | 48.5 | 23.6 |
| Qwen-Image | 85.6 | 61.9 | 79.8 | 79.2 | 84.3 | 86.1 | 58.6 | 40.5 | 35.8 | 18.2 | 86.6 | 16.3 | 83.5 | 82.1 | 39.3 |
| Qwen-Image-2512 | **88.0** | 91.7 | 82.7 | 82.6 | 91.1 | 92.5 | 60.8 | 38.2 | 37.7 | 20.9 | 88.3 | 21.6 | 88.7 | 88.0 | 45.0 |
| Z-Image-Turbo | 78.6 | 84.6 | 71.2 | 80.5 | 84.1 | 85.4 | 51.5 | 34.6 | 31.9 | 15.3 | 80.9 | 16.5 | 84.1 | 77.0 | 42.0 |
| LongCat-Image | 79.2 | 92.8 | 73.1 | 82.9 | 63.3 | 60.5 | 55.1 | 38.2 | 34.2 | 23.4 | 82.9 | 18.5 | 81.3 | 80.3 | **67.0** |
| **Autoregressive Models** | | | | | | | | | | | | | | | |
| Infinity-8B | 57.2 | 94.8 | 60.6 | 82.9 | 7.3 | 12.9 | 40.6 | 31.4 | 24.3 | 14.6 | 70.0 | 16.0 | 72.2 | 66.6 | 34.1 |
| GoT-R1-7B | 38.3 | 97.3 | 50.9 | 83.6 | 3.8 | 7.2 | 27.6 | 19.1 | 20.3 | 11.2 | 50.8 | 8.3 | 55.2 | 50.5 | 24.2 |
| **Unified Models** | | | | | | | | | | | | | | | |
| BAGEL | 58.2 | 96.4 | 62.5 | 83.6 | 4.3 | 11.3 | 30.1 | 22.3 | 22.0 | 11.0 | 65.3 | 8.6 | 58.0 | 40.3 | 24.3 |
| BAGEL w/ Think | 49.2 | 97.3 | 57.4 | 82.2 | 0.8 | 2.7 | 31.3 | 31.4 | 23.0 | 17.2 | 58.1 | 11.7 | 56.3 | 64.1 | 45.6 |
| show-o2-1.5B | 52.4 | 93.7 | 57.0 | 81.7 | 1.4 | 7.7 | 30.0 | 20.5 | 19.4 | 13.6 | 54.7 | 10.1 | 65.6 | 58.6 | 26.7 |
| show-o2-7B | 52.1 | 93.7 | 59.0 | 80.7 | 1.3 | 2.6 | 32.5 | 20.9 | 20.5 | 13.7 | 59.7 | 10.3 | 66.7 | 61.8 | 26.5 |
| Janus-Pro-1B | 41.5 | 95.7 | 50.0 | 82.3 | 1.7 | 3.4 | 15.4 | 14.1 | 11.9 | 10.7 | 48.3 | 5.4 | 42.6 | 8.3 | 6.3 |
| Janus-Pro-7B | 45.5 | 96.7 | 55.4 | 83.2 | 4.3 | 8.9 | 26.2 | 19.1 | 17.4 | 11.2 | 59.4 | 11.6 | 59.0 | 15.1 | 9.2 |
| BLIP3o-4B | 34.5 | 97.3 | 41.7 | 81.8 | 0.2 | 0.6 | 18.9 | 12.7 | 12.6 | 13.8 | 34.6 | 8.9 | 44.0 | 41.9 | 33.6 |
| BLIP3o-8B | 35.2 | 97.8 | 44.9 | 82.8 | 0.1 | 0.9 | 20.2 | 18.6 | 13.4 | 17.1 | 35.9 | 10.0 | 47.7 | 49.9 | 38.3 |
| OmniGen2-7B | 62.3 | 94.9 | 61.4 | 81.4 | 13.3 | 22.5 | 32.9 | 17.3 | 23.3 | 10.5 | 73.5 | 12.4 | 77.4 | 66.0 | 32.2 |
| HunyuanImage-3.0 | 83.3 | 94.0 | 81.1 | 82.7 | 84.4 | 85.3 | 54.5 | 32.7 | 36.4 | 19.8 | 82.2 | 26.1 | 79.8 | 78.4 | 37.3 |
| **Closed-Source Models** | | | | | | | | | | | | | | | |
| Seedream 3.0 | 82.2 | 68.4 | 76.9 | 84.3 | 39.0 | 52.1 | 49.1 | 37.7 | 31.8 | 21.0 | 83.2 | 19.6 | 84.6 | 81.6 | 36.4 |
| Seedream 4.0 | 91.0 | 94.0 | 84.5 | 84.9 | 95.2 | 92.6 | 82.2 | 82.7 | 74.4 | 45.9 | 86.6 | 42.6 | 80.5 | 90.9 | 82.2 |
| Seedream 4.5 | 90.9 | 97.6 | **87.8** | 82.6 | 97.8 | 94.6 | 86.8 | 81.4 | 64.4 | 50.2 | 90.4 | 47.8 | 82.3 | 91.9 | 84.8 |
| Gemini 2.0 Flash | 61.4 | 96.7 | 66.1 | 84.6 | 60.9 | 66.2 | 50.9 | 40.9 | 34.1 | 28.8 | 81.9 | 28.9 | 68.4 | 77.9 | 45.5 |
| Nano Banana | 83.0 | 98.7 | 76.5 | **87.6** | 84.7 | 87.4 | 83.9 | 77.7 | 57.3 | 60.2 | 83.1 | 50.6 | 84.7 | 88.6 | 80.6 |
| Nano Banana Pro | 90.8 | 97.8 | 85.2 | 86.1 | **98.7** | **97.9** | **92.3** | **91.8** | **91.1** | 69.8 | 88.9 | 66.9 | 88.9 | 91.6 | **89.3** |
| Imagen 4 | 81.0 | 92.7 | 72.7 | 85.6 | 90.4 | 85.8 | 60.6 | 46.4 | 38.3 | 42.9 | 84.1 | 34.5 | 81.8 | 84.6 | 75.0 |
| Imagen 4 Ultra | 88.4 | 97.7 | 79.1 | 86.9 | 85.6 | 85.8 | 81.3 | 70.9 | 55.2 | 56.6 | 84.1 | 51.7 | 82.1 | 86.6 | 79.6 |
| GPT-Image | 80.9 | **99.2** | 74.3 | 86.0 | 85.7 | 87.5 | 77.1 | 65.5 | 52.3 | 46.8 | 85.6 | 54.7 | 76.6 | 80.8 | 72.7 |
| GPT-Image-1.5 | 84.5 | 98.6 | 82.8 | 87.5 | 95.6 | 94.3 | 84.3 | 68.2 | 54.4 | 63.3 | 81.4 | 64.9 | 79.2 | 89.5 | 79.6 |

corresponds to cases that require a single step of logical inference (*e.g.*, "*the larger of two objects is on the left*"), whereas the **multi-hop (m-hop)** label corresponds to cases that require multiple chained inferences (*e.g.*, "*if the dog is behind the fence, and the fence is behind the house, then the dog is behind the house*").

- *BR* **Behavioral Reasoning:** The positive (**POS**) label is used to evaluate the model's core behavioral reasoning capability by verifying whether the image presents the inevitable visual consequences triggered by the behavior described in the prompt but not explicitly stated (*e.g.*, "*a glass is knocked over → the water spills onto the floor*"). In contrast, the negative (**NEG**) label is used to identify elements that remain unaffected by the behavior, preserving their original state (*e.g.*, "*knocking over a glass of orange juice does not affect the egg placed beside it*").

- **HR** *Hypothetical Reasoning:* The positive **(POS)** label is used to verify the visual results that directly follow from the hypothetical rule, where the corresponding objects satisfy the assumed premise and therefore should exhibit the specified change or characteristic (*e.g.*, "*if the wheels are assumed to be square, the car should display square wheels*"). Conversely, the negative **(NEG)** label is used to verify that objects not meeting the hypothetical premise remain unaffected, ensuring that the model does not mistakenly apply the hypothetical rule to inapplicable objects (*e.g.*, "*other parts of the car not mentioned in the hypothesis should remain unchanged*").

- **GR** *Generalization Reasoning:* The invariant **(INV)** label is used to evaluate features in the target scene that remain unchanged, representing the "common constant attributes" summarized across multiple examples (*e.g.*, "*all birds have wings*"). In contrast, the variant **(VAR)** label is used to assess whether the model can follow a cross-example variation logic to generate systematic changes in certain attributes within the target scene (*e.g.*, "*the color of each bird changes across different scenes while their shape remains the same*").

We report the fine-grained analyses in Table 6, and conclude the following interesting insights: **(1) Most models find NEG cases easier than POS, though a few notable exceptions emerge.** Across **MI**, **MA**, **BR**, and **HR**, models consistently score higher on NEG cases than POS ones, suggesting that it is generally easier to avoid conditions than to satisfy them. This trend is especially pronounced in reasoning tasks (**BR**, **HR**), where models are better at confirming the absence of change than at predicting correct outcomes. Interestingly, a few advanced models deviate from this pattern: both Qwen-Image and Seedream 3.0 slightly favors POS over NEG in **MI**, indicating their limitations in handling negative constraints. **(2) Performance on the two sub-dimensions of Text Rendering is strongly correlated, suggesting that both content and layout must be jointly optimized.** In the **TR** dimension, models that achieve high accuracy in textual content (CON) also tend to perform well in layout fidelity (LAY), and vice versa. This strong correlation implies that effective text rendering requires coordinated progress in both semantic correctness and spatial arrangement, as deficiencies in either aspect can significantly impair overall performance. **(3) A clear stepwise effect is observed in Logical Reasoning, with multi-hop problems being consistently more difficult than 0-hop/1-hop ones.** Across models, performance in **LR** declines noticeably as the number of reasoning hops increases, with multi-hop questions scoring lower than 1-hop, which in turn score lower than 0-hop. This pattern reflects the increasing complexity introduced by multi-step dependencies, indicating that current models struggle to maintain reasoning consistency over longer inferential chains. **(4) In Generalization Reasoning, models handle invariant patterns more reliably than variant ones.** Within the **GR** dimension, scores on the invariant subset (INV) are consistently higher than those on the variant subset (VAR). This indicates that models are more adept at identifying and preserving shared, stable patterns, but struggle when required to generalize over systematic variations. The performance gap reveals a core challenge in enabling models to reason beyond fixed regularities toward flexible pattern adaptation.

## C.3 QUANTITATIVE EXAMPLES AND COMPARISONS

Due to page limits, we include the complete set of illustrative examples and cross-model qualitative comparisons in Fig. 9 and Figs. 10, 11, 12. These figures showcase *composition* and three key dimensions of *reasoning* (*i.e.*, *deductive*, *inductive*, and *abductive*), providing a fuller picture beyond the main quantitative results in the text.

## D   LLM USAGE STATEMENT

In this work, LLMs are used solely as general-purpose assistive tools. Specifically, we use them to (1) provide suggestions for improving grammar and clarity of writing, (2) help organize section structures, and (3) assist in generating candidate prompts and checklists during the benchmark construction stage, which are subsequently verified and refined by human annotators. Importantly, all research ideas, experiment designs, and final scientific claims are developed and validated by the authors themselves. The LLMs do not contribute to the originality of research concepts or conclusions, and are therefore not considered contributors or co-authors. The authors take full responsibility for all content presented in this paper, including any text initially drafted with LLM assistance.

## E    LIMITATIONS AND DISCUSSION

**Limitations.** While our T2I-COREBENCH provides a comprehensive and challenging benchmark for assessing both compositional and reasoning capabilities, we also observe several limitations in evaluation: (i) Our study focuses solely on T2I generation, leaving out other emerging modalities such as video generation and interactive multimodal generation, which pose additional temporal and contextual reasoning challenges. (ii) Although our checklist-based evaluation ensures consistency and objectivity across dimensions, certain aspects could benefit from finer-grained metrics. For example, text rendering is currently assessed at the sentence level, whereas character-level accuracy could offer a more detailed perspective. (iii) Our benchmark primarily evaluates generative faithfulness with respect to prompt semantics, without considering non-semantic aspects such as aesthetics, realism, and diversity. The dataset largely focuses on objects and animals, with limited coverage of human-centric or face-related cases, which may reduce relevance to certain real-world applications. Expanding the benchmark to include human-related scenarios, together with broader non-semantic dimensions, is an important direction for future work. (iv) Our benchmark is currently limited to English prompts, while multilingual capabilities remain largely unexplored; extending the benchmark to multiple languages represents an important direction for future work.

**Discussion.** To address the identified challenges of T2I generation in complex composition and reasoning scenarios, we identify four promising research directions for future work: (i) The development of more diverse and challenging training data, particularly with multi-element and reasoning-oriented supervision, is essential for enabling stronger generalization across complex tasks. (ii) The integration of LLMs and MLLMs into T2I pipelines should be advanced, leveraging their strong language modeling and cross-modal reasoning capabilities to improve semantic understanding and alignment in complex generation scenarios. (iii) The incorporation of LLM-style reasoning paradigms, *e.g.*, Chain-of-Thought (Wei et al., 2022), Self-Consistency (Wang et al., 2022), and Retrieval-Augmented Generation (Gao et al., 2023), into T2I pipelines can facilitate intermediate inference before image generation, thereby improving the extraction of implicit visual elements from complex prompts. (iv) The exploration of reasoning mechanisms during generation is also needed, by explicitly integrating visual reasoning steps into the generation process to support more detailed and controllable outputs. We hope this benchmark and analysis can facilitate future research toward building T2I models into both "*set the stage*" and "*direct the play*".

***Multi-Instance (MI):*** A lively elementary school classroom during art period, where colorful student artwork decorates the walls above rows of small desks and bright yellow chairs. The teacher's desk holds a red apple, coffee mug, and scattered pencils, while a large whiteboard displays today's lesson plan written in blue marker. Near the windows, potted plants thrive on the windowsill next to boxes of tissues and hand sanitizer. Art supplies overflow from plastic bins: crayons, scissors, glue sticks, and construction paper in every imaginable color. However, you won't find any electronic tablets or computers in this traditional classroom, as the school maintains a hands-on learning approach. There are also no musical instruments like drums or guitars present, keeping the focus purely on visual arts. Students' backpacks hang on hooks along the back wall, while a globe sits prominently on a corner table beside stacks of picture books. The bulletin board showcases gold star stickers and student certificates, and alphabet letters march along the wall border above the green chalkboard. *35 Instances*

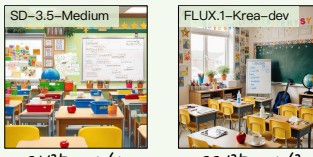 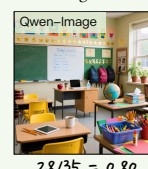 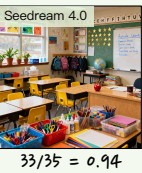 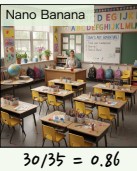 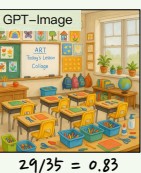

21/35 = 0.60    22/35 = 0.63    28/35 = 0.80    33/35 = 0.94    30/35 = 0.86    29/35 = 0.83

***Multi-Attribute (MA):*** A single mechanical clockwork dragon constructed from brass and copper gears in a Victorian inventor's workshop. The dragon is medium-sized with articulated joints and visible clockwork mechanisms throughout its body. Its scales are individual brass plates that overlap like medieval armor, and its eyes are glowing amber gemstones. The dragon has four legs with mechanical claws, and its wings are made of thin copper sheets with brass ribbing. It is not organic, being entirely mechanical in construction. Two steam vents are positioned along its spine, releasing small puffs of white vapor. The dragon's head features three rotating gear assemblies visible through transparent crystal panels. Its tail is segmented with spring-loaded joints and ends in a sharp brass spear point. The dragon's chest houses a large central clockwork heart that glows with warm golden light and produces visible ticking motion. Intricate engravings of Victorian flourishes decorate the brass surfaces, and tiny copper wires connect various mechanical components. The dragon is not corroded, maintaining its polished metallic appearance. Small brass screws and bolts are visible at every joint, and delicate filigree work adorns the wing membranes. The dragon shows no signs of rust and contains no modern electronic components. *25 Attributes*

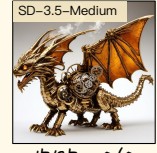 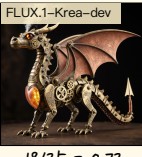 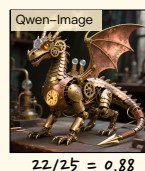 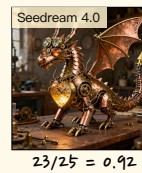 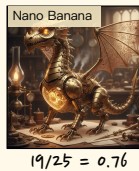 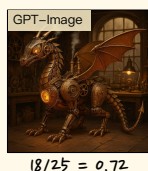

15/25 = 0.60    18/25 = 0.72    22/25 = 0.88    23/25 = 0.92    19/25 = 0.76    18/25 = 0.72

***Multi-Relation (MR):*** A kitchen scene with an experienced chef wearing a white hat standing behind an old, worn wooden counter. The chef is holding a large knife and cutting carrots on a cutting board. A shiny, new red pot sits on top of a gas stove next to the counter; the pot is noticeably newer than the counter. A black cat is sitting under the counter facing the chef. On the counter, there are three apples and a group of carrots. The number of carrots is twice the number of apples. The three apples are arranged in front of the cutting board. Also on the counter are some onions, and their number is one less than the number of apples. A wooden spoon is inside the red pot. The chef is pointing at a recipe book that lies open between the apples and the cutting board. The recipe book is thicker than the cutting board. A kitchen towel hangs from a hook on the wall behind the stove. A salt shaker sits next to the recipe book on the counter. *21 Relations *

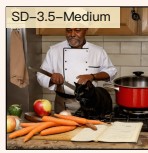 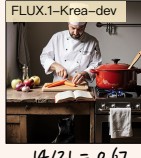 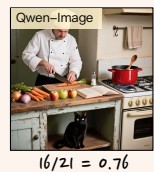 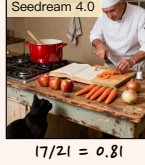 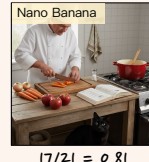 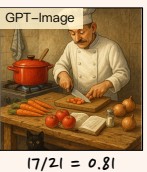

8/21 = 0.38    14/21 = 0.67    16/21 = 0.76    17/21 = 0.81    17/21 = 0.81    17/21 = 0.81

***Text-Rendering (TR):*** Create a pharmaceutical product packaging box with detailed multi-level text hierarchy. The main product name 'MEDIHEALTH PLUS™' should be displayed in large blue letters on the front panel. Below that, show 'Advanced Pain Relief Formula' in smaller black text. The package should have four information sections: 'ACTIVE INGREDIENTS' (top-left), 'DOSAGE INSTRUCTIONS' (top-right), 'WARNINGS & PRECAUTIONS' (bottom-left), and 'MANUFACTURER INFO' (bottom-right). Under ACTIVE INGREDIENTS, list 'Ibuprofen 400mg', 'Acetaminophen 325mg', and 'Caffeine 65mg'. Under DOSAGE INSTRUCTIONS, show 'Adults: 1-2 tablets', 'Every 6-8 hours', and 'Max: 6 tablets/day'. Under WARNINGS & PRECAUTIONS, display 'Do not exceed dosage', 'Consult doctor if pregnant', and 'Keep away from children'. Under MANUFACTURER INFO, list 'MediCorp International', 'Lot #: MH-2024-456', and 'Exp: 12/2026'. Add a small plus symbol (+) next to 'Ibuprofen 400mg' and 'Adults: 1-2 tablets' only. Do not add symbols next to any other text elements. On the side panel, include 'FDA APPROVED' and 'Store below 25℃'. *20 Texts + 20 Layouts *

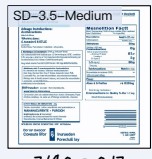 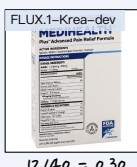 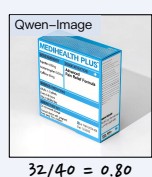 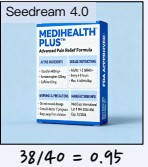 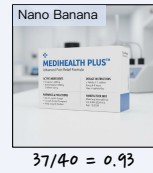 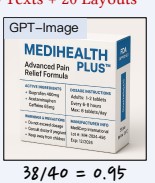

7/40 = 0.17    12/40 = 0.30    32/40 = 0.80    38/40 = 0.95    37/40 = 0.93    38/40 = 0.95

Figure 9: **Quantitative examples of *composition* dimensions (*i.e.*, MI , MA , MR , TR ).**

*Logical Reasoning (LR):* Generate an image of three robots in a lab. Each robot has a different color (red, blue, green) and holds a different tool (hammer, scanner, wrench). The robots make statements: (1) Red robot says: 'Blue robot has the hammer.' (2) Blue robot says: 'I have the scanner.' (3) Green robot says: 'Red robot is lying.' (4) Red robot also says: 'I have the wrench.' (5) Additional facts: Exactly one robot always lies, the other two always tell the truth. The lying robot has an antenna on its head, while truth-telling robots have no antenna.   *5 Premises*

---

*Checklist:*  01. Does the red robot have an antenna on its head?        04. Is the red robot holding the hammer?
02. Does the blue robot have no antenna on its head?     05. Is the blue robot holding the scanner?
03. Does the green robot have no antenna on its head?    06. Is the green robot holding the wrench?

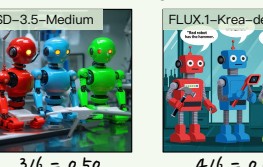 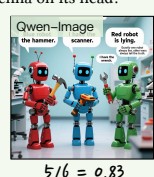 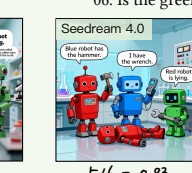 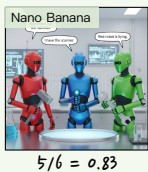 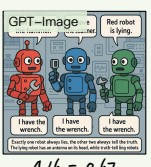

3/6 = 0.50      4/6 = 0.67      5/6 = 0.83      5/6 = 0.83      5/6 = 0.83      4/6 = 0.67

*Behavioral Reasoning (BR):* Generate a photo of a Rube Goldberg-style chain reaction in a classroom, captured at the final moment. The initial setup contains a taut elastic cord placed just before a line of standing dominoes, with a matchstick fixed at the midpoint of the cord under tension. Behind the domino line, the last domino is positioned to connect to a mechanism designed to cut the rope suspending a steel marble. The marble is aligned to roll down a ramp into a glass beaker filled with red-colored water, which rests on a white sheet of paper. Far to the side of this setup on the same desk is a closed microscope under a dust cover. The actions that just occurred: the matchstick is used to burn through the taut elastic cord, which, upon snapping, tips over the first domino in the line. The image should depict the scene after all resulting effects have completely finished.   *8 Outcomes*

---

*Checklist:*  01. Is the matchstick charred or blackened after burning?        05. Is the steel marble inside the glass beaker?
02. Is the elastic cord visibly broken after being burned through?   06. Is the white paper under the beaker stained with red splashes?
03. Are all of the dominoes in the line lying flat on the desk?      07. Is the microscope still on the desk, far from the experiment?
04. Does the rope holding the steel marble appear to be cut?         08. Is the dust cover still on the microscope and completely dry?

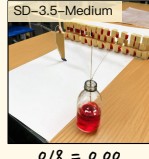 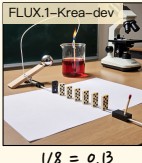 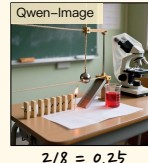 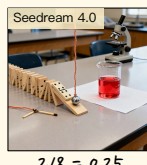 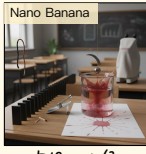 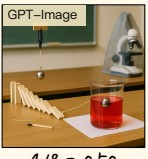

0/8 = 0.00      1/8 = 0.13      2/8 = 0.25      2/8 = 0.25      5/8 = 0.63      4/8 = 0.50

*Hypothetical Reasoning (HR):* Depict a bustling city-street intersection. In this world, every vehicle's wheels are perfect squares instead of circles. Present: 1) a yellow taxi car, 2) a red double-decker bus, 3) a delivery bicycle, 4) a police motorcycle, 5) a gray sedan, 6) a street-sweeper truck, 7) a pedestrian's shoes, 8) a street lamp, 9) a fire hydrant, 10) a public trash bin, 11) a blue mailbox, and 12) a traffic light pole. Render in daylight realism.   *12 Items*

---

*Checklist:*  01. Are the taxi's wheels depicted as perfect squares?               07. Do the pedestrian's shoes keep their normal soles?
02. Are the bus's wheels depicted as perfect squares?                08. Does the street lamp keep its normal form?
03. Are the bicycle wheels depicted as perfect squares?              09. Does the fire hydrant keep its normal form?
04. Are the motorcycle wheels depicted as perfect squares?           10. Does the trash bin keep its normal form?
05. Are the sedan's wheels depicted as perfect squares?              11. Does the mailbox keep its normal form?
06. Are the street-sweeper truck wheels depicted as perfect squares? 12. Does the traffic-light pole keep its normal form?

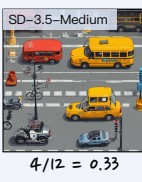 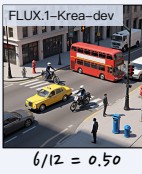 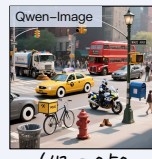 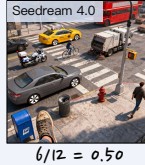 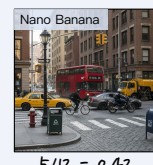 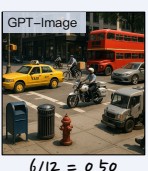

4/12 = 0.33      6/12 = 0.50      6/12 = 0.50      6/12 = 0.50      5/12 = 0.42      6/12 = 0.50

*Procedural Reasoning (PR):* Illustrate the final scene after performing all of the following six steps in order:\n1. Place a square sheet of purple origami paper flat on a wooden table.\n2. Fold the paper diagonally corner to corner and crease sharply, then unfold.\n3. Fold along the other diagonal and crease, then unfold to reveal an X-shaped crease pattern.\n4. Collapse the paper inward along the creases to form a square base.\n5. Continue folding to create the traditional bird base, then pull out the neck, head, and two wings to form a crane.\n6. Spread the wings gently so the crane stands upright and centre it on the table.\nRender the tabletop exactly as it appears once all six steps are complete.   *6 Procedures*

---

*Checklist:*  01. Is a finished origami crane made from purple paper present on the table?
02. Are both wings extended outward horizontally?        04. Is the crane's head distinct and bent slightly downward?
03. Is the crane standing upright without external support?   05. Is no unfolded sheet or scrap paper left on the table?

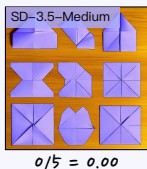 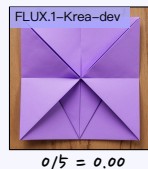 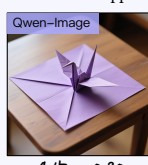 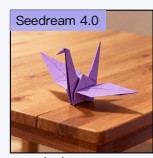 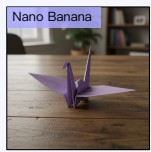 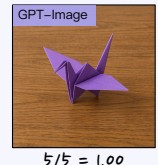

0/5 = 0.00      0/5 = 0.00      4/5 = 0.80      5/5 = 1.00      5/5 = 1.00      5/5 = 1.00

Figure 10: **Quantitative examples of *deductive reasoning* dimensions (*i.e.*, LR , BR , HR , PR ).**

**Generalization Reasoning (GR):** This is a system that creates a 'Divine Chariot' based on a 'Deity's Domain'. Study the examples to understand the rules.\n\nExample 1: The source is the 'Sky' domain, with 2 charioteers, a primary metal of gold, and a 'Sun' symbol. The result is a golden chariot that floats without wheels. It is pulled by 2 griffins (Rule: Sky -> Griffins/Floating, Sea -> Hippocampi/Wheels; creature count = charioteer count). A large, golden sun emblem is on the front of the chariot (Rule: metal determines chariot and emblem material). The chariot emits soft rays of light (Rule: Sun -> light rays, Trident -> water trails). The chariot is made of glowing energy and metal and is set against a cloudy sky.\n\nExample 2: The source is the 'Sea' domain, with 1 charioteer, a primary metal of silver, and a 'Trident' symbol. The result is a silver chariot with wheels made of swirling water. It is pulled by 1 hippocampus (Rule: Sky -> Griffins/Floating, Sea -> Hippocampi/Wheels; creature count = charioteer count). A large, silver trident emblem is on the front of the chariot (Rule: metal determines chariot and emblem material). The chariot is followed by trails of swirling water (Rule: Sun -> light rays, Trident -> water trails). The chariot is made of glowing energy and metal and is set against a stormy sea.\n\nNow, apply this exact system. Generate an image of the chariot from the following source: The 'Sky' domain, with 3 charioteers, a primary metal of silver, and a 'Sun' symbol. *8 Generalization Rules*

---

**Checklist:**
01. Is the chariot being pulled by griffins?
02. Are there exactly 3 griffins pulling the chariot?
03. Does the chariot float and have no wheels?
04. Is the body of the chariot made of silver?
05. Is there a large sun-shaped emblem on the front of the chariot?
06. Is the material of the emblem also silver?
07. Does the chariot emit soft rays of light?
08. Is the setting a sky with clouds?

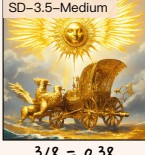 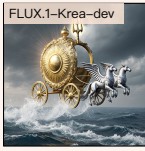 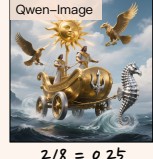 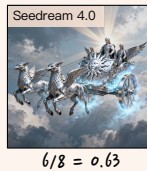 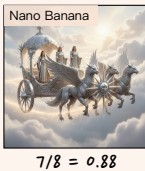 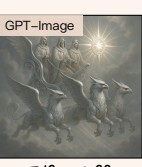

| SD–3.5–Medium | FLUX.1–Krea–dev | Qwen–Image | Seedream 4.0 | Nano Banana | GPT–Image |
|---|---|---|---|---|---|
| 3/8 = 0.38 | 3/8 = 0.38 | 2/8 = 0.25 | 6/8 = 0.63 | 7/8 = 0.88 | 7/8 = 0.88 |

**Analogical Reasoning (AR):** Just as a honeycomb displays the following visual properties: (1) each cell has exactly six sides, (2) all sides of each hexagon are the same length, (3) adjacent cells share common walls, (4) all hexagons are the same size, and (5) the hexagonal pattern covers the entire visible surface, create an image showing clouds arranged in the sky following this same organizational principle. The image should ultimately be guided by the visual analogy, prioritizing its rules over real-world physics. *5 Analogical Rules*

---

**Checklist:**
01. Does each cloud formation have exactly six sides?
02. Are all sides of each cloud hexagon the same length?
03. Do adjacent cloud formations share common walls?
04. Are all cloud hexagons the same size?
05. Does the hexagonal cloud pattern cover the entire visible sky area?

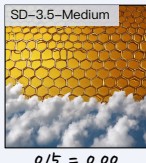 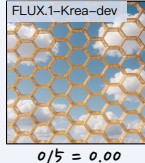 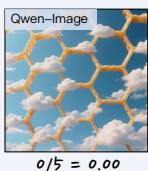 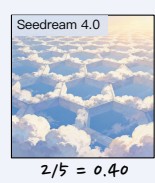 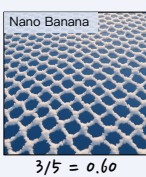 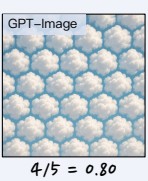

| SD-3.5–Medium | FLUX.1–Krea–dev | Qwen–Image | Seedream 4.0 | Nano Banana | GPT–Image |
|---|---|---|---|---|---|
| 0/5 = 0.00 | 0/5 = 0.00 | 0/5 = 0.00 | 2/5 = 0.40 | 3/5 = 0.60 | 4/5 = 0.80 |

Figure 11: **Quantitative examples of *inductive reasoning* dimensions (*i.e.*, GR , AR ).**

**Commonsense Reasoning (CR):** Describe a realistic scene, including any necessary real-world details to make it believable. In a veterinary clinic's examination room, a veterinarian is conducting a check-up on a nervous terrier. The vet is leaning over the animal to listen carefully to its heartbeat. The owner stands close by, stroking the dog's head to keep it calm. On the stainless steel counter in the background, a single syringe has been prepared next to a small vial. On the wall, an X-ray film is clipped onto an illuminated light box. *5 Commonsense*

---

**Checklist:**
01. Is the veterinarian using a stethoscope to listen to the dog's heartbeat?
02. Is the dog positioned on an elevated metal examination table?
03. Is the veterinarian wearing professional attire suitable for a medical environment, such as scrubs or a lab coat?
04. Is the needle of the prepared syringe on the counter still covered with its protective cap?
05. Does the illuminated X-ray on the wall display the skeletal structure of an animal?

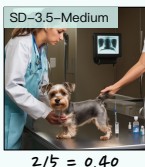 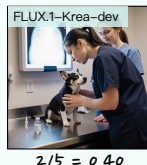 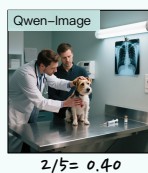 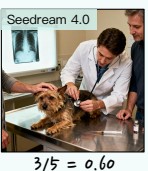 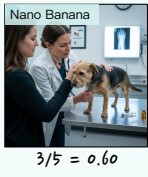 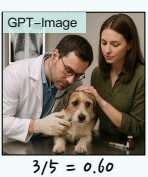

| SD–3.5–Medium | FLUX.1–Krea–dev | Qwen–Image | Seedream 4.0 | Nano Banana | GPT–Image |
|---|---|---|---|---|---|
| 2/5 = 0.40 | 2/5 = 0.40 | 2/5 = 0.40 | 3/5 = 0.60 | 3/5 = 0.60 | 3/5 = 0.60 |

**Reconstructive Reasoning (RR):** Observations:\nOn a bedroom windowsill sits an open jewelry box with one earring missing from a pair. A single black feather rests on the sill. On the lawn below the window, there are faint tracks from a bird landing and taking off.\n\nGenerative Task:\nReconstruct and generate a high-speed photograph of the precise and singular moment just after the theft has been completed, capturing the instant when the thief is about to escape. All objects mentioned in Observations must be reconstructed in the scene, except those that are meant to have disappeared in the reconstructed moment. *5 Clues*

---

**Checklist:**
01. Is a bird, such as a crow or magpie, visible in the scene?
02. Is the bird holding a shiny earring in its beak?
03. Is the bird depicted in mid-flight, taking off from the windowsill?
04. Is there an open jewelry box on the windowsill?
05. Is a single matching earring still visible inside the jewelry box?

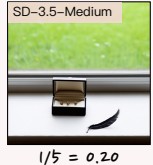 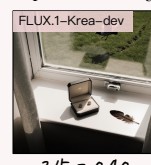 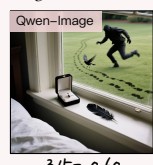 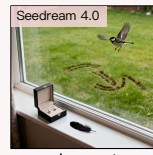 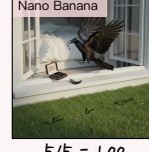 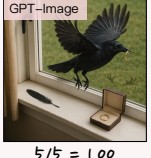

| SD–3.5–Medium | FLUX.1–Krea–dev | Qwen–Image | Seedream 4.0 | Nano Banana | GPT–Image |
|---|---|---|---|---|---|
| 1/5 = 0.20 | 2/5 = 0.40 | 3/5 = 0.60 | 2/5 = 0.40 | 5/5 = 1.00 | 5/5 = 1.00 |

Figure 12: **Quantitative examples of *abductive reasoning* dimensions (*i.e.*, CR , RR ).**

