# OpenReview forum: "Easier Painting Than Thinking: Can Text-to-Image Models Set the Stage, but Not Direct the Play?"
_ICLR.cc/2026/Conference — ICLR 2026 Poster_

### Official Review · Reviewer_DGHz · 2025-10-26

**Soundness:** 3
**Presentation:** 3
**Contribution:** 3
**Rating:** 6
**Confidence:** 4

**Summary:**

The authors introduce T2I-CoReBench, a benchmark for evaluating both composition and reasoning capabilities of text-to-image (T2I) generation models with 12 dimensions. The benchmark comprises 1,080 prompts and ~13,500 checklist visual questions, enabling fine-grained evaluation with an MLLM-based yes/no answerer. Experiments on 28 T2I models (diffusion, autoregressive, and unified architectures) reveal that while composition is steadily improving, reasoning remains the main performance bottleneck. Further, prompt rewriting shows some effectiveness in bridging this gap but remains limited in scenarios requiring deeper visual reasoning.

**Strengths:**

1. The authors provides the first benchmark that evaluate both composition and reasoning capabilities of T2I models.

2. The main findings (reasoning lags behind composition & prompt rewriting helps but still there’s gap) is well supported with large experiments (e.g., table 3) and valuable for future research direction.

**Weaknesses:**

1. **Evaluation method.** In L143-145, the authors mentioned `We propose an automatic checklist-based evaluation protocol ... individual yes/no questions ... allows fine-grained and reliable assessment' as a second contribution. However, such automatic checklist-based evaluation protocol (i.e., question generation followed by question answering) was already extensively studied in previous works [e.g., A, B, C, D], but none of them have been cited or discussed. I don't think authors can claim evaluation methodological novelty, which is separate from benchmark construction. The authors should cite these works and clarify their contributions.

- [A] Hu et al., TIFA: Accurate and Interpretable Text‑to‑Image Faithfulness Evaluation with Question Answering. ICCV 2023
- [B] Yarom et al., What You See is What You Read? Improving Text‑Image Alignment Evaluation. NeurIPS 2023.
- [C] Cho et al., Visual Programming for Text‑to‑Image Generation and Evaluation. NeurIPS 2023
- [D] Cho et al. Davidsonian Scene Graph: Improving Reliability in Fine‑grained Evaluation for Text‑to‑Image Generation. ICLR 2024.

**Questions:**

1. This is minor point, but I feel the title is hard to understand. I didn't understand what title meant before reading the paper, and didn't expect to read text-to-image evaluation benchmark paper at all. Now I guess that authors meant composition and reasoning by 'set the stage' and 'direct the play', but still, such connection wouldn't be clear for many new readers.

---

> ### Author Response · Authors · 2025-11-21
> **Author Response**
>
> > W1: Evaluation method. In L143-145, the authors mentioned `We propose an automatic checklist-based evaluation protocol ... individual yes/no questions ... allows fine-grained and reliable assessment' as a second contribution. However, such automatic checklist-based evaluation protocol (i.e., question generation followed by question answering) was already extensively studied in previous works [e.g., A, B, C, D], but none of them have been cited or discussed. I don't think authors can claim evaluation methodological novelty, which is separate from benchmark construction. The authors should cite these works and clarify their contributions.
> >
>
> We sincerely thank the reviewer for this important observation and apologize for the misleading wording in our original submission. We fully acknowledge that the idea of using an automatic checklist-based evaluation protocol has been thoroughly studied in prior works [A, B, C, D]. It was not our intention to imply methodological novelty in this evaluation paradigm, and we recognize that our phrasing incorrectly suggested otherwise.
>
> Our intended contribution is not the evaluation protocol itself, but rather the **benchmark we constructed**, which includes a large-scale set of around 13,500 **human-verified, atomic, dimension-specific checklist questions** covering both explicit compositional elements and implicit reasoning outcomes. The contribution lies in the **scope, scale, and human verification** of this checklist as part of the benchmark. We appreciate the reviewer for pointing out this oversight. Following the reviewer's suggestion, we have revised the corresponding phrasing in the main text as follows:
>
> |  | Original | Revised |
> | --- | --- | --- |
> | Line 144-146 | We propose an automatic checklist-based evaluation protocol aligned with human judgments, consisting of around $13,500$ individual \textit{yes/no} questions in total. This allows fine-grained and reliable assessment of whether the generated images capture both explicit and implicit visual elements. | **We pair each prompt with a human-verified checklist of individual \textit{yes/no} questions, for a total of around $13,500$ questions across the benchmark.** This facilitates fine-grained and reliable assessment of whether the generated images faithfully capture both explicit and implicit elements. |
> | line 324-328 | In this regard, we pair each prompt with a checklist of independent \textit{yes/no} questions (with the correct answer always ``\textit{Yes}'’), enabling fine-grained verification of both explicit and implicit elements. | **To facilitate fine-grained and reliable evaluation of both explicit and implicit visual elements, we follow previous visual-question-answering–based evaluation paradigms~\cite{hu2023tifa, yarom2023you, cho2023visual, cho2023davidsonian},** by pairing each prompt with a checklist of independent \textit{yes/no} questions (with the correct answer always ``\textit{Yes}'’). |
>
> ---
>
> > Q1: This is minor point, but I feel the title is hard to understand. I didn't understand what title meant before reading the paper, and didn't expect to read text-to-image evaluation benchmark paper at all. Now I guess that authors meant composition and reasoning by 'set the stage' and 'direct the play', but still, such connection wouldn't be clear for many new readers.
> >
>
> We thank the reviewer for the helpful feedback. Our intention with the metaphorical title was to express, in an illustrative way, the phenomenon we observe in current text-to-image models. Specifically, these models are often able to produce the overall scene composition, which corresponds to “set the stage,” but they frequently struggle with the implicit reasoning required to generate coherent outcomes, which corresponds to “direct the play.”
>
> We will also adopt a more direct name (e.g., *Easier Painting Than Thinking: A Comprehensive Benchmark for Complex Composition and Reasoning in Text-to-Image Models*) for our future project repository and benchmark webpage, so that readers can immediately grasp the focus and usage of our benchmark.

---

> > ### Comment · Reviewer_DGHz · 2025-11-21
> >
> > I thank the authors for their response and corresponding updates. I maintain my positive rating of the paper.

---

> > > ### Author Response · Authors · 2025-11-23
> > >
> > > We sincerely thank the reviewer for the positive feedback. Your comments have been very helpful in improving the rigor and clarity of our paper. We are glad that the revisions addressed your concerns and appreciate your continued support.

---

### Official Review · Reviewer_jqux · 2025-10-28

**Soundness:** 2
**Presentation:** 3
**Contribution:** 3
**Rating:** 6
**Confidence:** 3

**Summary:**

This paper introduces T2I-CoReBench, a Composition and Reasoning Benchmark for systematic evaluation of T2I models. T2I-CoReBench addresses research gaps in comprehensiveness and complexity, evaluating both composition and reasoning within a broader taxonomy. The dataset is constructed by an automatic pipeline, using LRMs to generate data points given instructions, and further examined by humans. Each data point is accompanied by a checklist, utilizing Google Gemini Flash 2.5 as the evaluation protocol to assess checklist accomplishment and provide a quality evaluation score. Comprehensive experiments evaluate 28 T2I models with different architectures and find that most models fall short on overall performance.

**Strengths:**

1. Solid contribution in constructing a comprehensive and challenging benchmark on T2I. T2I-CoReBench comprehensively covers composition and reasoning tasks in a detailed taxonomy.
2. Broad evaluation of different models offers great insights: The paper benchmarks 28 T2I models, covering open-source and closed-source models, as well as different architectures (Diffusion, Autoregressive, Unified). This makes the results comprehensive and convincing. Insights on "Prompt Rewriting" in Section 4.3 are intriguing.
3. Good Presentation Quality.

**Weaknesses:**

1. Evaluation relies on a single indicator, which cannot comprehensively measure the quality of T2I: There is only one indicator, the achievement rate of checklist questions, which cannot measure such as the overall rationality and stylization of the pictures. Given the examples, it also mainly examines item appearance in images, regardless of model hallucinations, etc.
2. Both the composition and reasoning tasks have their own previous benchmarks. Merely merging the two tasks into a more comprehensive benchmark limits their necessity.

**Questions:**

See weaknesses above.

---

> ### Author Response · Authors · 2025-11-21
> **Author Response (Part 1/2)**
>
> > W1: Evaluation relies on a single indicator, which cannot comprehensively measure the quality of T2I: There is only one indicator, the achievement rate of checklist questions, which cannot measure such as the overall rationality and stylization of the pictures. Given the examples, it also mainly examines item appearance in images, regardless of model hallucinations, etc.
> >
>
> Thanks for your comment. We agree that holistic indicators such as overall rationality and aesthetics are also important for evaluating T2I models, where we have already explicitly acknowledged this limitation in Appx. E. In this work, however, we design the evaluation protocol out of the following considerations:
>
> **(1) Our benchmark mainly focuses on semantic faithfulness, which we view as the most fundamental requirement for T2I models.** Semantic faithfulness remains insufficiently assessed in existing work, especially when models must handle both explicit descriptions and implicit inferences in complex, high-density scenes. To address this gap, we design our benchmark around comprehensiveness and complexity, covering both composition and reasoning across 12 dimensions with substantially richer visual density and multi-step inference challenges. As summarized in Table 1, this offers broader and deeper coverage than prior benchmarks, which typically evaluate only partial capabilities or operate in simpler scenarios.
>
> **(2) Our evaluation based on fine-grained checklist items still reflects overall rationality at the scene level.** Many items jointly encode global constraints that require coherence across the entire image. **In composition dimensions,** quantities, attribute bindings, and spatial relations must mutually align, so any irrational configuration will violate the corresponding checks and reduce the overall score. **In reasoning dimensions,** this coupling is even stronger because each expected outcome depends on a coordinated set of causal, hypothetical, or procedural dependencies. A single incorrect inference often disrupts multiple downstream conditions, revealing inconsistency across the entire reasoning chain. For example, in Behavioral Reasoning (BR), the snapped cord, fallen dominoes, and final position of the marble are interlocked consequences of the same process; if the model fails to infer any part of this chain, several related checks fail together. Through these structured dependencies, the evaluation captures whether the model maintains global rationality while still offering interpretable, fine-grained signals that pinpoint where the breakdown occurs.
>
> **(3) Our checklist-based evaluation protocol can also capture model hallucinations across both composition and reasoning tasks.** Appearance errors of models reveal **composition hallucinations**. Through MI, MA, MR, and TR, we explicitly specify which objects, attributes, relations, and text elements must or must not appear, so any hallucination, such as a spurious addition or a missing required element, will be detected. Similarly, **reasoning hallucinations** are revealed in BR, HR, PR, and RR, which encode the visual outcomes required by causal, hypothetical, procedural, and commonsense inferences. If a model hallucinates an outcome that is not supported by these inferences, it will fail multiple checklist items tied to the expected outcome. Through this structure, the benchmark can uncover both composition hallucinations and reasoning hallucinations of models.
>
> **(4) We acknowledge that broader overall indicators such as realism and aesthetic quality are also important, but they are orthogonal to our goal.** These aspects capture non-semantic properties of generation and often correlate weakly with factual correctness. Since our benchmark specifically targets semantic faithfulness, we intentionally keep these dimensions separate to avoid conflating visual style with semantic accuracy. That said, incorporating complementary metrics on visual realism and aesthetic quality is a valuable direction, and we view this as meaningful future work building on top of our benchmark.

---

> ### Author Response · Authors · 2025-11-21
> **Author Response (Part 2/2)**
>
> > W2: Both the composition and reasoning tasks have their own previous benchmarks. Merely merging the two tasks into a more comprehensive benchmark limits their necessity.
> >
>
> Thank you for the comment. We respectfully clarify that our benchmark is not a simple merge of composition and reasoning tasks. As summarized in Table 1, our core contribution lies in simultaneously addressing **comprehensiveness** and **complexity**, two limitations that existing benchmarks do not resolve even when considered together.
>
> **(1) Comprehensiveness: Existing benchmarks only cover isolated subsets of capabilities, whereas ours provides a unified and principled 12-dimension taxonomy.** Prior composition benchmarks typically evaluate only instances, attributes, or relations, while reasoning benchmarks focus on a single type of inference (e.g., commonsense, behavioral, or logical). None of them provides systematic coverage across both capabilities or across the full space of deductive, inductive, and abductive reasoning. In contrast, our taxonomy is grounded in scene-graph theory and philosophical inference frameworks, yielding a structured evaluation over all essential compositional elements and eight distinct reasoning types. Table 1 clearly shows large coverage gaps in existing work, which our benchmark fills in a unified and principled way.
>
> **(2) Complexity: We substantially increase visual density and inference intensity beyond all prior benchmarks.** Even within individual task types, previous datasets typically remain at low scene density (≤ 5 elements) or single-step reasoning (one behavior → one outcome). Our benchmark introduces high-density composition (≈ 20–35 elements per prompt) and multi-step reasoning (one-to-many and many-to-one inference), which reflect real-world generation demands and expose failure modes that prior benchmarks cannot reveal. Table 1 highlights this distinction: our benchmark is the only one that provides high complexity across *all* dimensions, whereas earlier benchmarks provide high complexity only in isolated slices, if at all.
>
> In summary, our goal is not merely to combine two existing task families, but to provide the first benchmark that is simultaneously **comprehensive in capability coverage** and **challenging in real-world complexity**. This unified and principled design is necessary for evaluating modern T2I models that are expected to operate under realistic semantic and reasoning conditions.

---

> > ### Comment · Reviewer_jqux · 2025-11-26
> >
> > I thank the authors for their response. I recognize this work's contribution and will maintain my positive rating.

---

> > > ### Author Response · Authors · 2025-11-27
> > >
> > > We sincerely thank the reviewer for the follow-up comment and truly appreciate your recognition of our contribution to building a comprehensive and complex T2I evaluation benchmark. We are grateful for your positive support of our work.

---

### Official Review · Reviewer_UbDT · 2025-11-01

**Soundness:** 3
**Presentation:** 3
**Contribution:** 3
**Rating:** 4
**Confidence:** 3

**Summary:**

The paper presents T2I-CoReBench, a benchmark for evaluating the compositional and reasoning abilities of text-to-image (T2I) models. The benchmark dissects model performance across multiple reasoning dimensions—logical, behavioral, hypothetical, and generalization reasoning—along with compositional skills such as multi-instance, multi-attribute, and text rendering. The authors argue that current T2I models excel in simple composition but struggle with structured reasoning that connects textual causality and visual outcomes. Comprehensive experiments compare state-of-the-art open- and closed-source models, uncovering systematic reasoning failures.

**Strengths:**

1. The authors categorize reasoning into fine-grained dimensions (e.g., logical, behavioral, hypothetical, abductive), offering a structured framework that is more interpretable than single-score benchmarks.

2. The inclusion of over a dozen models, spanning proprietary and open-source variants, makes the findings broadly representative and informative.

**Weaknesses:**

1. Some reasoning categories overlap (e.g., behavioral vs. hypothetical), raising questions about independence and orthogonality among evaluation axes.

**Questions:**

1. Is there any bias in the prompt generated synthetically?
2. Is there any distinction between the generated images of open and closed source models?

---

> ### Author Response · Authors · 2025-11-21
> **Author Response (Part 1/2)**
>
> > W1: Some reasoning categories overlap (e.g., behavioral vs. hypothetical), raising questions about independence and orthogonality among evaluation axes.
> >
>
> Thank you for raising this important point. We agree that ensuring conceptual clarity across reasoning dimensions is crucial. To guarantee orthogonality, we make efforts at both the theoretical and practical levels.
>
> - **Theoretically**, our categorization is grounded in the established tripartite framework of deductive, inductive, and abductive inference [1,2,3], and we further operationalize this structure into eight reasoning dimensions, each emphasizes a distinct inference mechanism (e.g., behavior-to outcome-causality for BR, counterfactual propagation for HR, etc.) that yields non-overlapped reasoning requirements.
> - **Practically**, our data construction pipeline is explicitly designed to maintain this orthogonality. Each dimension has its distinct Task Goal, Prompt Design Guidelines, and Checklist Construction Rules as described in Appx. A.1 to A.3. In addition, every sample is verified through multiple rounds of human checking, where annotators ensure that both the prompt and the checklist strictly align with the intended dimension and do not introduce elements from other reasoning types.
>
> Regarding the specific example of Behavioral Reasoning (BR) versus Hypothetical Reasoning (HR), we would like to clarify the difference in their data-construction and inference processes:
>
> - Behavioral Reasoning (BR) focuses on real-world causal processes: the prompt specifies an initial physical setup and one or more actions, and the model must infer the inevitable chain of consequences that unfold in the real world (e.g., burning a cord → cord snaps → dominoes fall → marble drops). The core challenge is predicting deterministic physical outcomes driven by actual behaviors.
> - Hypothetical Reasoning (HR) centers on counterfactual premises: the prompt introduces a rule that contradicts real-world physics(e.g., “every vehicle’s wheels are perfect squares in this world”), and the model must uniformly enforce this counterfactual rule on the affected objects, while preserving normal appearance for unaffected ones. The core challenge is to enforce a global counterfactual rule across the scene, instead of predicting real-world physical outcomes in BR.
>
> To help clarify the distinctions among the dimensions in the main paper, we use Table 2 to summarize each dimension side-by-side with its definition and complexity specification, and use Fig. 3 to provide visualized examples that illustrate how the dimensions manifest in concrete T2I scenarios. In the revised version, we highlight in the main text (line 247-250) the above discussion over our categorization orthogonality and refer readers to Appx. A for more detailed definitions and construction criteria of all 12 dimensions.
>
> [1] Peirce, Charles Sanders. *Collected papers of charles sanders peirce*. Vol. 5. Harvard University Press, 1934.
>
> [2] Nodelman, Uri, Colin Allen, and John Perry. "Stanford encyclopedia of philosophy." 2003.
>
> [3] Godfrey-Smith, Peter. "Theory and reality: An introduction to the philosophy of science." *Theory and reality*. University of Chicago Press, 2009.

---

> ### Author Response · Authors · 2025-11-21
> **Author Response (Part 2/2)**
>
> > Q1: Is there any bias in the prompt generated synthetically?
> >
>
> Thank you for the thoughtful question. Since our prompts are initially generated by LRMs, we fully acknowledge the risk of model-induced bias. In practice, we observe two major forms of bias during data construction: **(i) intra-model stylistic bias internal to each LRM**, and **(ii) cross-model content imbalance and redundancy**. To mitigate these biases, we adopt two complementary strategies:
>
> **(1) Using three different LRMs to reduce intra-model stylistic bias.** As detailed in Sec. 3.3, all prompts and checklists are jointly generated by **three different SOTA LRMs** (Claude Sonnet 4, Gemini 2.5 Pro, and OpenAI o3). Considering that each LRM comes with inherent stylistic preferences (e.g., specific syntactic patterns, typical entity orderings, and recurring narrative structures), using a single model alone would cause its style to dominate the dataset. By drawing samples from three distinct models for every dimension, we substantially diversify linguistic expression and prevent the benchmark from inheriting any single LRM’s stylistic or structural tendencies.
>
> **(2) Multi-round human verification and refinement to reduce cross-model content bias.** After initial generation, each LRM produces around 100 candidate samples per dimension, which then undergo multi-stage human verification and refinement (Appx. A.2 and A.3). Annotators refine the outputs by correcting unclear or inconsistent descriptions, removing redundancy, and ensuring strict adherence to the dimension-specific guidelines. Crucially, during this filtering stage, annotators also remove overly similar or template-like cases across LRMs, ensuring that the final 30-per-LRM set reflects balanced thematic and structural diversity rather than accidental concentration on a few favored scene types.
>
> For example, in the Commonsense Reasoning (CR) dimension, all three LRMs propose scenarios involving the Mexican Día de los Muertos festival; after human review, only one representative sample was retained, preventing the final dataset from being unintentionally skewed toward that specific cultural theme.
>
> ---
>
> > Q2: Is there any distinction between the generated images of open and closed source models?
> >
>
> Thank you for the question. We would like to note that our paper already provides **extensive quantitative and qualitative evidence** comparing open- and closed-source models. Quantitatively, this includes (1) the Main Results in Sec. 4.2, (2) the effect of Prompt Rewriting in Sec. 4.3, and (3) the Fine-Grained Analysis in Appx. C.3. Qualitatively, Appx. C.4 provides cross-model visual comparisons across all 12 dimensions. Below, we provide a brief summary based on these results.
>
> **Quantitative summary.** Closed-source models show more stable performance across both composition and reasoning. For instance, Seedream 4.0 achieves 86.1 in composition and 69.9 in reasoning, and Imagen 4 Ultra reaches 82.4 and 72.9, maintaining relatively strong results even in reasoning-intensive dimensions such as HR, BR, and RR. Open-source models, while improving quickly, exhibit a sharper gap between explicit and implicit tasks: Qwen-Image scores 78.0 in composition but drops to 49.3 in reasoning, and diffusion models such as FLUX.1-Krea-dev (56.0 → 44.3) and SD-3.5-Medium (41.0 → 33.9) show similar declines. These results, consistent with Sec. 4.2, Sec. 4.3, and Appx. C.3, highlight that closed-source models handle multi-step inference more reliably, whereas open-source models perform best on explicit compositional grounding.
>
> **Qualitative illustration.** Figs. 9–12 provide qualitative comparisons across all 12 dimensions. Overall, closed-source models generally produce images that include more explictly or implicitly specified objects with accurate attributes and spatial relations, whereas open-source diffusion models more frequently omit objects, misrender attributes, or fail to follow the described structure. Take Text Rendering (TR) as an example, closed-source models such as Seedream 4.0, Nano Banana, and GPT-Image generate clearer text content and maintain coherent layout structure, whereas open-source diffusion models like SD-3.5-Medium and FLUX.1-dev frequently exhibit misspellings, collapsed characters, and inconsistent placement of titles or section headers. These qualitative observations align with the quantitative results, where the same closed-source models achieve the highest TR scores and the same diffusion-based models show substantial gaps.

---

### Official Review · Reviewer_pbo7 · 2025-11-01

**Soundness:** 3
**Presentation:** 3
**Contribution:** 3
**Rating:** 8
**Confidence:** 3

**Summary:**

This paper introduces T2I-COREBENCH, a high-density, checklist-based benchmark to jointly test composition (multi-instance, multi-attribute, multi-relation, text rendering) and reasoning (eight subtypes derived from deductive/inductive/abductive patterns) for text-to-image models. Evaluating 28 recent models, the authors find a clear gap: models can “paint” the specified elements but struggle to “think” through multi-step or implicit requirements, especially in dense scenes. Reasoning, not composition, is the current bottleneck.

**Strengths:**

- Well-motivated split between composition vs. reasoning, with 12 concrete dimensions.
- Dense prompts + yes/no checklists enable fine-grained, scalable evaluation.
- Broad study on 28 models gives useful community signals about where T2I still fails.

**Weaknesses:**

- Evaluator dependence and possible bias. The pipeline crucially depends on an MLLM to judge success. The paper would be stronger with a multi-evaluator or human-heavy subset to rule out evaluator-specific artifacts .

**Questions:**

- How did you de-correlate prompt/checklist style from the evaluator to avoid agreement-by-style?
- Did you experiment with soft scores or multiple MLLM votes, and if not, why stick to strict yes/no, which can make dense prompts disproportionately harsh?

---

> ### Author Response · Authors · 2025-11-21
> **Author Response (Part 1/2)**
>
> > W1: Evaluator dependence and possible bias. The pipeline crucially depends on an MLLM to judge success. The paper would be stronger with a multi-evaluator or human-heavy subset to rule out evaluator-specific artifacts
> >
>
> We thank the reviewer for highlighting this concern. We address it from two perspectives below:
>
> **(1) Human alignment study:**  During benchmark construction, we have conducted a dedicated human–MLLM consistency study to select an MLLM as evaluator (Appx. B.2, Table 5). We compared multiple candidate evaluators against human annotators.  As shown in Table 5, Gemini 2.5 Flash demonstrates strong agreement with humans and reliably preserves the same preference trends across dimensions, indicating that it is well aligned with human judgments for the types of visual questions in our benchmark.
>
> While a human-heavy subset might further reduce evaluator-specific artifacts, it would also introduce variability and fairness concerns, since different groups would rely on different annotators when reusing our benchmark. Given the strong empirical alignment between the MLLM and human judgments and the need for a reusable, scalable evaluation pipeline, we would recommend the MLLM evaluator as a practical substitute.
>
> **(2) Multi-evaluator evaluation:** Besides our main results with Gemini-2.5-Pro (Sec. 4.2, Table 3), we additionally report results using two open-source SOTA MLLMs: Qwen2.5-VL-72B (Appx. C.2, Table 6) and Qwen3-VL-30B-Thinking (Appx. C.2, Table 7). We observe that all three evaluators yield highly consistent performance trends. For example, the top-3 models remain essentially consistent, with Qwen-Image, FLUX.1-Krea-dev, and Infinity-8B leading under open-source evaluators, and Imagen 4 Ultra, Nano Banana, and Seedream 4.0 leading under closed-source evaluators.
>
> Inspired by your advice, we explore a multi-evaluator fusion strategy, where a checklist item is counted as “yes” only if **all three MLLMs** (Gemini-2.5-Flash, Qwen2.5-VL-72B, and Qwen3-VL-30B-Thinking)  predict “yes”. We include the results in Table 8 (Appx. C.3) of the revised Appendix, where its results exhibit consistent performance trends with those reported in the main paper (Table 3), with the open-source top-3 remaining Qwen-Image, FLUX.1-Krea-dev, and Infinity-8B, and the closed-source top-3 remaining Imagen 4 Ultra, Nano Banana, and Seedream 4.0.
>
> Our original intention in using an MLLM as the evaluator was to keep the benchmark easy to adopt, making it convenient for others to reuse our benchmark for their own assessments. In our project repository, we will also provide the fused multi-MLLM evaluator option, so that the committee and future users can choose the evaluation configuration that best fits their needs.

---

> ### Author Response · Authors · 2025-11-21
> **Author Response (Part 2/2)**
>
> > Q1: How did you de-correlate prompt/checklist style from the evaluator to avoid agreement-by-style?
> >
>
> We appreciate the reviewer’s question. To alleviate this agreement-by-style, we have taken the following strategies during benchmark construction:
>
> **(1) Prompts and checklists are generated by three different LRMs.** As emphasized in Sec. 3.3, we employ three SOTA LRMs (i.e., Claude Sonnet 4, Gemini 2.5 Pro, and OpenAI o3) during data construction to mitigate stylistic homogeneity and potential bias arising from relying on a single LRM. This significantly diversifies linguistic style and prevents the evaluator from benefiting from same-model or same-family phrasing.
>
> **(2) All samples undergo multi-round human verification and refinement.** After generating the initial samples with LRMs, all data undergoes multi-stage human filtering and refinement (Appx. A.2 and A.3), including feasibility checking, redundancy removal, and iterative human verification. This human-in-the-loop process naturally mitigates model-induced biases by correcting stylistic artifacts and filtering out LRM-specific tendencies.
>
> **(3) The evaluator is instructed not to rely on the original textual prompt for assessment solely.** During evaluation, the evaluator is required to answer each checklist question based on the visual content of the generated image. As specified in the evaluation instruction (Appx. B.2, Fig. 7),  given the human-verified checklist, a “yes” response is permitted only when the required visual element is clearly present in the image. This design prevents the evaluator from relying on linguistic priors or prompt phrasing, thereby alleviating style-driven shortcuts and ensuring strictly image-grounded assessment.
>
> **(4) Multi-evaluator cross-checking to alleviate stylistic bias.** We additionally evaluate with Qwen2.5-VL-72B and Qwen3-VL-30B-Thinking, whose trends match Gemini 2.5 Flash closely. We also expect the **multi-evaluator fusion strategy** explored above to validate this consistency further.
>
> ---
>
> > Q2: Did you experiment with soft scores or multiple MLLM votes, and if not, why stick to strict yes/no, which can make dense prompts disproportionately harsh?
> >
>
> We appreciate the reviewer’s insightful question. Our choice of strict yes/no scoring follows two considerations:
>
> **(1) The nature of our semantic evaluation necessitates binary decisions.** Our benchmark is designed on **semantic faithfulness**, where each checklist item is an atomic visual question (Sec. 3.2). Many such questions are inherently binary—for example, “Is there an apple in the image?” A soft score such as 0.5 is not well-defined in this context: does it mean the apple is partially visible, vaguely apple-like, or simply ambiguous? Such interpretations vary across evaluators and undermine interpretability.
>
> **(2) Harshness is intentional to reflect real-world compositional and reasoning failures.** Since our benchmark is designed to evaluate T2I models under real-world complexities (e.g., dense prompts), the scoring must be sensitive enough to expose weaknesses that only appear in such settings. In dense prompts, each visual element represents a distinct reasoning or compositional requirement, and missing even a minor component often indicates a substantive failure. A stricter rule therefore prevents partial correctness and yields a more faithful and discriminative measure of true capability in high-complexity cases.
>
> **We agree that using multiple MLLM votes can provide a “multi-expert’’ alternative.** Using several MLLMs can be viewed as consulting multiple independent experts, which offers a broader evaluation perspective. In our experiments (W1 above), the aggregated votes and single-MLLM evaluation exhibit very similar performance trends and model rankings. Given that a benchmark should remain lightweight and easy for users to adopt, we adopt the single-MLLM setting (Gemini-2.5-Pro) as default in the main paper. At the same time, to ensure robustness and reproducibility, we also report results from two more MLLMs (Qwen2.5-VL-72B and Qwen3-VL-30B-Thinking) in Appx. C.2.

---

> > ### Comment · Reviewer_pbo7 · 2025-11-27
> >
> > Thank you for your response; this has resolved my concerns.

---

### Author Response · Authors · 2025-11-21
**Summary of Paper Revision**

We gratefully thank all the reviewers for their valuable and constructive comments. We are encouraged that they find our motivation to split composition and reasoning clear and well-motivated (`pbo7`), our taxonomy comprehensive and fine-grained (`UbDT` and `jqux`), and our checklist-based evaluation fine-grained and interpretable (`pbo7`). We are also glad that they acknowledge our extensive experiments on 28 models provide useful community insights (`pbo7` and `jqux`) and that our main findings are well-supported and valuable for future research directions (`DGHz`).

We address the concerns and questions in detail below. According to these comments, we have also improved our manuscript (modifications are marked in **red**) and summarized the main changes as follows:

1. **Supplementary experiments with multiple evaluators (Appx. C.2 & C.3, to `pbo7`):** We added evaluation results using two more open-source MLLMs (Qwen2.5-VL-72B and Qwen3-VL-30B-Thinking) and a multi-evaluator fusion strategy, demonstrating that the current evaluation trends remain robust across different evaluators.
2. **More discussion on orthogonality over evaluation dimensions (Sec. 3.1 & Appx. A.4, to `UbDT`):** We highlighted the orthogonality of our evaluation dimensions in the main text from both theoretical and practical perspectives.
3. **Clarification of contributions and related work (Sec. 1 & 3.2, to `DGHz`):** We revised the phrasing regarding the automatic checklist-based evaluation protocol to properly cite prior works (e.g., TIFA, Davidsonian Scene Graph, etc.) and clarified that our contribution focuses on the construction of our benchmark with the large-scale, human-verified checklist-based assessment.

---

### Author Response · Authors · 2025-12-03
**General Response and Summary of Discussion**

We sincerely thank the Area Chair and all reviewers for their time and constructive feedback. We are encouraged that **3 out of 4 reviewers (`pbo7`, `jqux`, `DGHz`)** **explicitly confirmed their satisfaction with our rebuttal and maintained their positive ratings.** Regarding Reviewer `UbDT`, although they have not responded to our rebuttal, we believe our detailed clarifications on dimension orthogonality and bias mitigation have effectively addressed their concerns.

During the discussion period, we actively engaged with all reviewers to address concerns regarding evaluation robustness, taxonomy clarity, and methodological positioning. Below is a summary of the discussion.

| **Reviewer** | **Rating** | **Confidence** | **Strengths** | **Questions** | **Our Reply** | **Feedback** |
| --- | --- | --- | --- | --- | --- | --- |
| **`pbo7`** | 8: Accept | 3: Fairly Confident | **(1)** **Well-motivated** split between composition and reasoning. **(2)** Dense prompts with **fine-grained** evaluation. **(3)** **Broad study** on 28 models and **useful insights**. | **(1) Evaluator Bias:** Dependence on single MLLM in evaluation. **(2) Style Bias:** De-correlate prompt/checklist style biases from evaluators. **(3) Scoring:** Is strict yes/no scoring too harsh? | **(1)** We explained that our evaluation is highly aligned with humans through human alignment study and provided multi-evaluator evaluation results (including a fusion strategy) to demonstrate the robustness of our assessment. **(2)** We clarified that we use 3 different LRMs with multi-round human verification **during generation** and instruct the evaluator to focus on visual evidence only **during evaluation.** **(3)** We justified that strict binary scoring could avoid ambiguity and effectively reflects real-world failures in dense scenes. | **Overall Satisfied** ("Thank you for your response; this has resolved my concerns.") |
| **`UbDT`** | 4: BR | 3: Fairly Confident | **(1)** **Fine-grained** reasoning dimensions. **(2)** **Structured** and more **interpretable** framework. **(3)** **Broad evaluation** with representative and informative findings. | **(1) Orthogonality:** Overlap between dimensions (e.g., Behavioral vs. Hypothetical). **(2) Prompt Bias:** Potential bias in synthetic prompts. **(3) Comparisons:** Distinction between open vs. closed source images. | **(1)** We clarified dimension orthogonality by distinguishing **theoretical** reasoning mechanisms (e.g., real-world causality for Behavioral vs. counterfactual justification for Hypothetical) and **practical** construction pipeline that enforces strict separation via guidelines and human verification. **(2)** We explained that we use 3 distinct LRMs combined with multi-round human verification to minimize stylistic and content bias. **(3)** We provided detailed discussion with quantitative (Sec 4.2) and qualitative (Appx C.4) evidence to demonstrate the  differences between images generated by open and closed-source models. | **No response** (Reviewer did not reply to our rebuttal.) |
| **`jqux`** | 6: BA | 3: Fairly Confident | **(1)** **Solid contribution** in constructing a comprehensive and challenging benchmark. **(2)** **Broad evaluation** with great insights. **(3)** **Good presentation quality**. | **(1) Metric:** Single indicator for item appearance may neglect model hallucinations. **(2)** **Comparisons:** Contribution beyond merely merging previous benchmarks. | **(1)** We clarified that this benchmark uses atomic checklists to evaluate fundamental semantic faithfulness for T2I generation, and our evaluation involving complex reasoning can also reflect overall rationality and model hallucinations. **(2)** We compared previous benchamrks with ours in Table 1, which emphasizes our contribution lies in both **comprehensivenss** (*i.e.*, 12-dimension taxonomy) and **complexity** (*i.e.*, visual density and inference intensity). It resolves the two key limitations of previous benchmarks, namely their narrow capability coverage and low task complexity, which cannot be addressed even if those benchmarks are merged. | **Overall Satisfied** ("I recognize this work's contribution and will maintain my positive rating.") |
| **`DGHz`** | 6: BA | 4: Confident | **(1)** The **first** benchmark evaluating both composition and reasoning in T2I. **(2)** **Large experiments with valuable findings** for future research direction. | **(1) Citations:** Missing citations for VQA-based protocols (TIFA, DSG). **(2) Title Expression:** Metaphorical title hard to understand. | **(1)** We revised the text to explicitly cite TIFA and DSG etc., clarifying that our contribution is the large-scale benchmark data construction rather than the VQA protocol itself. **(2)** We explained the title's metaphor and agreed to add a descriptive subtitle to the project repository for immediate clarity. | **Overall Satisfied** ("I maintain my positive rating of the paper.") |

---

### Meta-Review · Area_Chair_gyfF · 2026-01-04

**Summary:**

This paper presents a new benchmark called T2I-COREBENCH, to evaluate the scene composition and (deductive, inductive, and abductive) reasoning abilities of T2I models with 1080 prompts and 12 metrics. The benchmark also has a human-verified checklist of 13500 questions to allow more reliable evaluation.

The reviewers shared several concerns, including evaluator bias, reasoning category overlap, bias in prompts, lack of related work discussions. 3 out of 4 reviewers acknowledged that the authors properly addressed their concerns. The reviewer that gave score 4 did not participate in discussions, but I believe they would have raised the score if they were able to discuss with the authors.

I recommend acceptance given that this paper makes a solid contribution to the comprehensive evaluation of T2I models' abilities, and most of the reviewers hold positive opinions about this work. Please incorporate all reviewers' suggestions in the next version of the paper.

Please correct the citations and use \citep and \citet properly. Throughout the paper, the majority of the citations were used incorrectly. You should use \citep{} to add parentheses to the citations when they are not to be read as a part of the sentence.

**Reviewer Concerns:**

Reviewer pbo7 gave a short review and raised concerns about evaluator bias since the pipeline heavily depends on MLLM. The authors addressed the concern with human study and multi-evaluators. The reviewer acknowledged that the concerns were addressed.

Reviewer UbDT gave a very short review and raised concern about reasoning category overlap and bias in prompts, and the authors properly addressed. The reviewer did not participate in the author-reviewer discussion. I believe the reviewer would have raised the score if they were to participate.

Reviewer jqux gave a short review and raised concerns about the single indicator which is a limitation that the authors acknowledged and provided detailed explanations for. Another concern was that there exist previous benchmarks, and the authors clarified the differences. I think those concerns are properly addressed. The reviewer acknowledged the concerns were addressed.

Reviewer DGHz gave a relatively detailed review, and raised concerns about the eval method and lack of related work discussion. The reviewer acknowledged the concerns were addressed.

**Reviewer Scores:**

See above.

---

### Decision · Program_Chairs · 2026-01-26

Accept (Poster)